# Reprogramming and redifferentiation of mucosal-associated invariant T cells reveal tumor inhibitory activity

Chie Sugimoto[1], Yukie Murakami[1], Eisuke Ishii[2], Hiroyoshi Fujita[1], Hiroshi Wakao[1]*

[1]Host Defense Division, Research Center for Advanced Medical Science, Dokkyo Medical University, Mibu, Japan; [2]Department of Dermatology, School of Medicine, Dokkyo Medical University, Mibu, Japan

**Abstract** Mucosal-associated invariant T (MAIT) cells belong to a family of innate-like T cells that bridge innate and adaptive immunities. Although MAIT cells have been implicated in tumor immunity, it currently remains unclear whether they function as tumor-promoting or inhibitory cells. Therefore, we herein used induced pluripotent stem cell (iPSC) technology to investigate this issue. Murine MAIT cells were reprogrammed into iPSCs and redifferentiated towards MAIT-like cells (m-reMAIT cells). m-reMAIT cells were activated by an agonist in the presence and absence of antigen-presenting cells and MR1-tetramer, a reagent to detect MAIT cells. This activation accompanied protein tyrosine phosphorylation and the production of T helper (Th)1, Th2, and Th17 cytokines and inflammatory chemokines. Upon adoptive transfer, m-reMAIT cells migrated to different organs with maturation in mice. Furthermore, m-reMAIT cells inhibited tumor growth in the lung metastasis model and prolonged mouse survival upon tumor inoculation through the NK cell-mediated reinforcement of cytolytic activity. Collectively, the present results demonstrated the utility and role of m-reMAIT cells in tumor immunity and provide insights into the function of MAIT cells in immunity.

## Editor's evaluation

This study exploits a reprogramming approach to study MAIT cells that can partially overcome the current technological limitations. Overall, the reviewers found that this work is provocative and novel.

*For correspondence:
hwakao@dokkyomed.ac.jp

Competing interest: The authors declare that no competing interests exist.

## Introduction

Mucosal-associated invariant T (MAIT) cells belong to a family of innate-like T cells harboring semi-invariant T-cell receptors (TCRs) and recognize vitamin B2 metabolites as antigens on major histocompatibility complex class I-related gene protein (MR1) (*Godfrey et al., 2019*). MAIT cells play a pivotal role in immunity by bridging innate and adaptive immunities. They are abundant in humans, but rare in mice, and are associated with a number of diseases, such as bacterial and viral infections, autoimmune, inflammatory, and metabolic diseases, asthma, and cancer (*Godfrey et al., 2015*; *Godfrey et al., 2019*; *Toubal et al., 2019*). Although previous studies implicated MAIT cells in various tumors, their role in tumor immunity remains obscure. An analysis of infiltrating immune cells revealed that the high infiltration of CD8+CD161+ T cells, largely comprising MAIT cells, type 17 CD8+ T cells (Tc17 cells), and stem cell-like memory cells, in tumors represented a favorable prognostic signature across a wide array of human cancers (*Gentles et al., 2015*; *Konduri et al., 2021*). In mucosal-associated cancers, including lung and colon cancers, increases in MAIT cells in tumors were concomitant with a decrease in the circulation (*Ling et al., 2016*; *Sundström et al., 2015*; *Won et al., 2016*). While the

high infiltration of MAIT cells in tumors negatively correlated with a favorable outcome in colorectal cancer (CRC) patients, the low infiltration of these cells in the tumors of hepatocellular carcinoma patients correlated with a poor prognosis, which is in contrast to the role of MAIT cells in tumor immunity (*Zabijak et al., 2015*; *Zheng et al., 2017*). Furthermore, the tumor microenvironment endowed by tumor-infiltrating lymphocytes, tumor-associated myeloid cells, and cancer-associated fibroblast plays a pivotal role in cancer progression and regression. Accordingly, tumor-residing MAIT cells exhibit an activated and/or exhausted phenotype as exemplified by the upregulation of CD39, PD-1, and *CXCL13* upon the stimulation of TCR (*Leng et al., 2019*; *Shaler et al., 2017*; *Yao et al., 2020*). Consistent with these findings, tumor-infiltrating MAIT cells from CRC show the decreased production of cytokines that are pertinent to antitumor activity (*Ling et al., 2016*; *Shaler et al., 2017*; *Sundström et al., 2015*). Nevertheless, it currently remains unclear whether these changes positively or negatively affect the function of MAIT cells in tumor immunity (*Cogswell et al., 2021*). However, a recent study based on single-cell RNA sequencing revealed that approximately 50% of Tc17 cells in tumor-infiltrating T cells comprised MAIT cells across various tumors, and these Tc17 cells were particularly enriched in hepatocellular carcinoma and cholangiocarcinoma (*Zheng et al., 2021*). In addition, the frequency of Tc17 cells in melanoma patients correlated with responsiveness to anti-PD-1 therapy (*Sade-Feldman et al., 2019*; *Zheng et al., 2021*). The latter findings argue the antitumor activity of MAIT cells.

Based on these discrepancies, further studies are warranted to clarify the role of MAIT cells in tumor immunity in mice. However, difficulties are still associated with examining MAIT cells in mice for the following reasons. These cells are 10- to 100-fold less abundant in mice than in humans. Although the advent of MR1-tetramer (MR1-tet) has enabled the detection of murine MAIT cells, the paucity of these cells per se still hampers functional studies (*Corbett et al., 2014*; *Rahimpour et al., 2015*). Furthermore, there is a lack of appropriate mouse models to investigate the function of MAIT cells (*Garner et al., 2018*; *Godfrey et al., 2019*; *Toubal et al., 2019*). Although the development of MAIT cells is dependent on MR1, MR1-deficient mice do not necessarily reveal all aspects of MAIT cells because they are also devoid of other MR1-dependent cells and provide only limited information on immune cells that potentially interact with MAIT cells. Similarly, previous studies on MAIT cell-specific TCR transgenic mice revealed the protective roles of MAIT cells in bacterial infections, type I diabetes, and experimental autoimmune encephalitis (*Chua et al., 2011*; *Croxford et al., 2006*; *Cui et al., 2015*; *Le Bourhis et al., 2010*; *Martin et al., 2009*; *Meierovics et al., 2013*; *Reantragoon et al., 2013*; *Sakala et al., 2015*; *Shalapour et al., 2012*; *Shimamura et al., 2011*). However, the aberrant expression of transcription factors in MAIT cells impedes investigations on their precise roles in health and disease.

We herein used induced pluripotent stem cell (iPSC) technology to examine the role(s) of MAIT cells in immunity, particularly tumor immunity. We reprogrammed murine MAIT cells into iPSCs (referred to as MAIT-iPSCs), differentiated MAIT-iPSCs into MAIT-like cells (referred to as m-reMAIT cells), and adoptively transferred them into syngeneic immunocompetent mice to investigate their role. This approach not only allows us to assess the role of m-reMAIT cells as effector cells, it also provides insights into the interactions with other immune cells in tumor immunity.

In this study, we showed that 5-(2-oxopropylideneamino)-6-D-ribitylaminouracil (5-OP-RU), an agonist of MAIT cells, and 5-OP-RU-loaded murine MR1-tet (mMR1-tet) both activated m-reMAIT cells, while these reagents induced the production of a similar, but not identical, set of cytokines and chemokines in the absence of antigen-presenting cells (APCs). When adoptively transferred, m-reMAIT cells migrated to different organs concomitant with the acquisition of maturation and prolonged mouse survival upon tumor inoculation. This tumor inhibitory activity was mediated through enhancements in the cytolytic activity of m-reMAIT cells by NK cells.

The present results imply that the adoptive transfer of m-reMAIT cells has potential as a novel tool for examining the immune functions of MAIT cells in health and disease.

## Results

### MAIT cell reprogramming and redifferentiation/characterization of m-reMAIT cells

Fluorescent-activated cell sorting (FACS)-purified MAIT cells (defined as CD44$^+$TCRβ$^+$mMR1-tet$^+$ cells, purity >97%) from the lungs of C57BL/6 mice were reprogrammed with the Sendai virus vector KOSM302L (*Wakao et al., 2013*). Forty-six iPSCs were obtained with a reprogramming efficiency of 0.6%. Specific iPSCs were subjected to limiting dilutions, and clone L7-1 was used throughout the experiments unless otherwise indicated. Since TCR comprises α and β chains, the presence of rearranged *Trav1-Traj33* specific for MAIT cells was detected by PCR followed by DNA sequencing and a Southern blot analysis with MAIT cell-derived iPSCs (MAIT-iPSCs) (*Figure 1—figure supplement 1A–D*). MAIT-iPSCs also harbored rearranged *Trbv13-3*, *Trbv19*, and *Trbv12-1* (*Table 1*) and exhibited pluripotency, as evidenced by the ability to give rise to chimeric mice (*Figure 1—figure supplement 1E*).

The differentiation of MAIT-iPSCs on OP9 ectopically expressing delta-like 1 (OP9-DLL1), a feeder cell for T-cell differentiation, resulted in m-reMAIT cells that were TCRβ$^+$mMR1-tet$^+$CD8$^+$CD4$^+$CD44$^-$ CD25$^{int}$PLZF$^-$RORγt$^+$ (*Figure 1A*). Consistent with this result, other MAIT-iPSC clones, such as L3-1, L11-1, L15-1, and L19-1, showed similar expression profiles to m-reMAIT cells on day 23 (*Figure 1— figure supplement 1H*). m-reMAIT cells multiplied more than 100-fold starting from MAIT-iPSCs with a doubling time of ~10.5 hr during the logarithmic phase of growth later in differentiation (*Figure 1— figure supplement 1F and G*). m-reMAIT cells were activated in a manner that was dependent on both 5-OP-RU and MR1 in the presence of APCs, such as WT3, the mouse embryonic fibroblast cell line, WT3/mMR1, WT3-overexpressing murine MR1, CH27, a B1B lymphoma cell line expressing a low level of endogenous MR1 on the cell surface, and CH27/mMR1, CH27-overexpressing murine MR1 (*Figure 1B*), which was consistent with previous findings (*Cheng et al., 1999*; *Huang et al., 2008*; *Miley et al., 2003*). Similarly, 5-OP-RU dose-dependent activation was observed in the other m-reMAIT cell lines L3-1, L11-1, L15-1, and L19-1 (*Figure 1—figure supplement 1I*). The activation of m-reMAIT cells in the presence of CH27 or CH27/mMR1 in turn induced the production of a battery of inflammatory cytokines and chemokines, such as IFN-γ, TNF-α, IL-17A, CCL4 (MIP-1β), and CCL5 (RANTES), as well as IL-2 and the T helper (Th)2 cytokine IL-13, in a 5-OP-RU dose-dependent manner (*Figure 1—figure supplement 1J* ). 5-OP-RU and mMR1-tet each activated m-reMAIT cells and promoted the production of a similar, but not identical, set of cytokines and chemokines in a dose-dependent manner in the absence of APCs. It is important to note that mMR1-tet was superior to 5-OP-RU at inducing TNF-α, Th2 cytokines, such as IL-6, IL-10, and IL-13, the Th17 cytokines, IL-17F, IL-22, and IL-23, and inflammatory chemokines, including CCL5, CCL2 (MCP-1), CXCL1, and CXCL6, irrespective of their similar abilities to upregulate CD69 (*Figure 1D–F*).

Although a TCR stimulation elicits an array of signaling cascades, including protein tyrosine kinases, phosphatases, GTP-binding proteins, and adaptor proteins, we herein focused on whether 5-OP-RU activated protein tyrosine kinases and induced tyrosine phosphorylation. The results obtained showed the dose- and time-dependent phosphorylation of proteins (*Figure 1G and H*). Linker for the activation of T cells (LAT) was identified as a TCR signaling component among phosphorylated proteins (*Figure 1I*). Unexpectedly, the addition of mMR1-tet, a reagent widely used to detect MAIT cells, engendered similar protein tyrosine phosphorylation, as was the case for 5-OP-RU (*Figure 1G, H and J*).

These results demonstrated that MAIT-iPSCs gave rise to m-reMAIT cells upon differentiation under T-cell permissive conditions, and m-reMAIT cells were activated by 5-OP-RU and mMR1-tet in the absence of APCs concomitant with protein tyrosine phosphorylation and the production of cytokines and chemokines.

### m-reMAIT cell migration into different organs

To follow the dynamics of m-reMAIT cells, cells were adoptively transferred into mice and their migration and maturation status were examined. To distinguish donor cells from endogenous MAIT cells, C57BL/6 (Ly5.1) mice received m-reMAIT cells that were Ly5.2. m-reMAIT cells and endogenous MAIT cells were detected in the thymus, bone marrow, lung, liver, spleen, intestines, and the mediastinal and inguinal lymph nodes at various endogenous MAIT cell/m-reMAIT cell ratios (*Figure 2A and D*).

**Table 1.** T-cell receptor β (TCRβ) repertoires of the mouse mucosal-associated invariant T (MAIT) cell-derived induced pluripotent stem cells (iPSCs).

| iPS clones | Congenic strain | Vβ | D | Jβ | V region end | V-D junction | D region | D-J junction | J region start | Nucleotide sequence | Translation |
|---|---|---|---|---|---|---|---|---|---|---|---|
| | | | | | | | | | | CDR3 | |
| | | | | | | V-D-J junction | | | | | |
| L3-1 | | TRBV13-3*01 | TRBD1*01 | TRBJ2-3*01 | TGATG | CTA | GGGACAGGG | | GTGCA | GCCAGCAGTGATGCTAGGGACAGGGGTGCAGAAACGCTGTAT | ASSDARDRGAETLY |
| L7-1 | | TRBV13-3*01 | TRBD1*01 | TRBJ1-2*01 | AGTGA | | CAGGG | A | AAACT | GCCAGCAGTGACAGGGAAAACTCCGACTACACC | ASSDRENSDYT |
| L11-1 | Ly5.2 | TRBV19*01, *03 | TRBD2*01 | TRBJ2-3*01 | GCAGT | | GGGACTGGGGGG | T | AGTGC | GCCAGCAGTGGACTGGGGGGTAGTGCAGAAACGCTGTAT | ASSGLGGSAETLY |
| L15-1 | | TRBV13-3*01 | TRBD1*01 | TRBJ1-6*01 | AGCAG | CA | GACAGGG | | CTATA | GCCAGCAGCAGACAGGGCTATAATTCGCCCCTCTAC | ASSRQGYNSPLY |
| L19-1 | | TRBV19*01, *03 | TRBD2*01 | TRBJ2-3*01 | GCAGT | | GGGACTGGGGGG | T | AGTGC | GCCAGCAGTGGACTGGGGGGTAGTGCAGAAACGCTGTAT | ASSGLGGSAETLY |

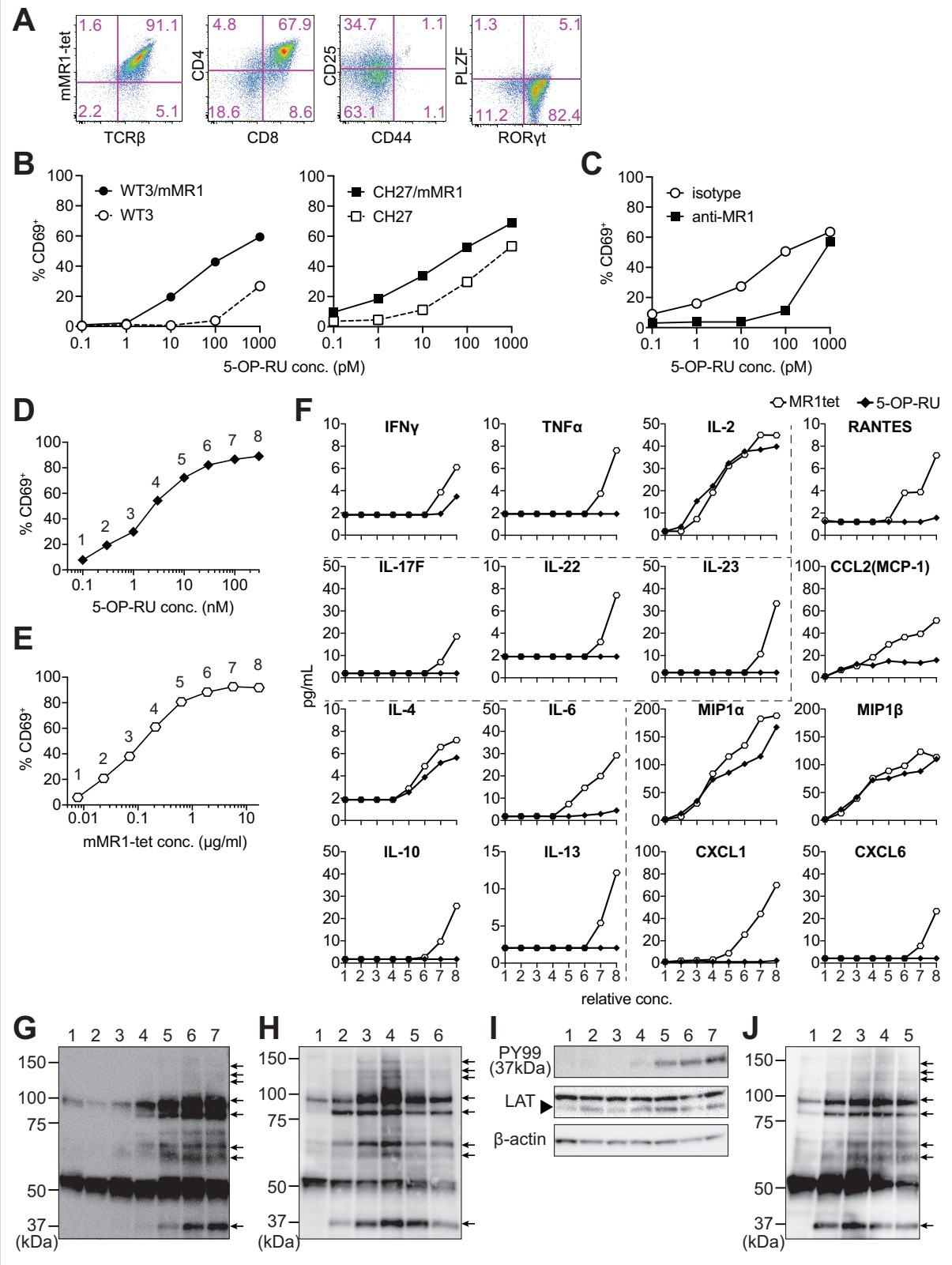

**Figure 1.** Characterization of m-reMAIT cells. (**A**) Flow cytometric profiles of m-reMAIT cells. mMR1-tet staining and expression of T-cell receptor β (TCRβ), CD4, CD8, CD25, and CD44 and the transcription factors PLZF and RORγt in m-reMAIT cells on differentiation day 18. (**B**) 5-(2-oxopropylideneamino)-6-D-ribitylaminouracil (5-OP-RU) dose-dependent activation of m-reMAIT cells. The percentages of CD69+ cells among m-reMAIT cells challenged with the indicated concentration of 5-OP-RU in the presence of WT3 (○), WT3/mMR1 (●), CH27 (□), and CH27/mMR1(■).

*Figure 1 continued on next page*

*Figure 1 continued*

(**C**) MR1-dependent activation of m-reMAIT cells. The percentage of CD69+ cells among m-reMAIT cells cultured with CH27/mMR1, challenged as in (**B**) in the presence of the anti-MR1 antibody (■) or the isotype control antibody (○). (**D**) 5-OP-RU dose-dependent activation. The percentage of m-reMAIT cells expressing CD69 upon a challenge with various concentrations of 5-OP-RU. Representative data from two independent experiments are shown. (**E**) mMR1-tet dose-dependent activation. The percentage of m-reMAIT cells expressing CD69 upon a challenge with the indicated amounts of mMR1-tet. Representative data from two independent experiments are shown. (**F**) 5-OP-RU- and mMR1-tet-induced cytokines and chemokines. m-reMAIT cells were stimulated with various concentrations of 5-OP-RU (○) or mMR1-tet (■) and the resultant cytokines and chemokines were quantified with LegendPlex. The concentrations at which each reagent induced a similar degree of activation (% CD69) are shown as relative concentrations (0.1–100 nM for 5-OP-RU and 0.01–10 μg/ml for mMR1-tet). The number on the X-axis corresponds to that in (**D**) and (**E**). (**G**) Tyrosine phosphorylation elicited with 5-OP-RU. A Western blot analysis with PY99 (anti-phosphotyrosine). Upon a challenge with different concentrations of 5-OP-RU for 30 min, the cell lysate from m-reMAIT cells was separated on SDS-PAGE ($5 \times 10^5$/lane), and subjected to Western blotting. Lane 1, 0; lane 2, 0.1; lane 3, 1.0; lane 4, 10; lane 5, 100; lane 6, 1000; and lane 7, 10,000 (nM). Phosphorylated proteins are indicated with arrows. (**H**) Time course of tyrosine phosphorylation. A Western blot analysis with PY99. The cell lysate from m-reMAIT cells challenged with 100 nM of 5-OP-RU for the indicated time was separated on SDS-PAGE ($5 \times 10^5$/lane) and subjected to Western blotting. Lane 1, 0; lane 2, 15; lane 3, 30; lane 4, 60; lane 5, 150; lane 6, 300 (min) . Arrows indicate phosphorylated proteins. (**I**) Linker for the activation of T cells (LAT) as a phosphorylated 37-kD protein. A Western blot analysis with PY99, anti-LAT, and anti-β-actin. The cell lysate prepared as described in (**H**) for 60 min was subjected to Western blotting. A blot with PY99 (upper panel), anti-LAT (middle panel), and anti-β-actin (lower panel). Phosphorylated LAT and LAT as well as β-actin are indicated (arrow). Lane 1, 0; lane 2, 0.1; lane 3, 1.0; lane 4, 10; lane 5, 100; lane 6, 1000; lane 7, 10,000 (nM). (**J**) Tyrosine phosphorylation induced by mMR1-tet. A Western blot analysis with PY99. The cell lysate from m-reMAIT cells challenged with the indicated amounts of unlabeled mMR1-tet for 60 min was subjected to Western blotting ($5 \times 10^5$/lane). Lane 1, 0; lane 2, 0.43; lane 3, 1.3; lane 4, 4.3; lane 5, 13 (μg/ml). Arrows indicate phosphorylated proteins. MAIT: mucosal-associated invariant T cell.

The online version of this article includes the following source data and figure supplement(s) for figure 1:

**Source data 1.** Characterization of m-reMAIT cells.

**Source data 2.** Original gel electrophoresis panels for *Figure 1G–J*.

**Figure supplement 1.** Characterization of induced pluripotent stem cells (iPSCs) from mucosal-associated invariant T (MAIT) cells and of m-reMAIT cells.

**Figure supplement 1—source data 1.** Characterization of induced pluripotent stem cells (iPSCs) from mucosal-associated invariant T (MAIT) cells and of m-reMAIT cells.

**Figure supplement 1—source data 2.** Gel electrophoresis for PCR products and Southern blot analysis to detect *Trav1-Traj33* rearrangement.

Since CD44 levels reflect the degree of the functional maturation of T cells (*Budd et al., 1987*), we examined time-dependent changes in CD44 expression in m-reMAIT cells. In comparison with endogenous MAIT cells, m-reMAIT cells showed the time-dependent upregulation of CD44 in the spleen, liver, and lungs, and lower expression in the intestines (*Figure 2B*).

An analysis of other molecules relevant to the functionality of MAIT cells revealed quasi-equivalent IL-18Rα expression in the spleen, liver, and intestines to that in endogenous cells. Moreover, the expression of IL-7Rα and CXCR6 in all organs examined was lower than that in endogenous cells, while CD69 expression was quasi-equivalent to that in endogenous cells (*Figure 2C*). Furthermore, the frequency of m-reMAIT cells among mMR1-tet+TCRβ+ cells in primary lymphoid organs decreased, while that in peripheral tissues increased over time (*Figure 2D*).

We then investigated the transcriptome in naïve m-reMAIT cells, m-reMAIT cells adoptively transferred into syngeneic mice, and endogenous MAIT cells from the recipient mice to follow the evolution of transcripts by RNA sequencing. A principal component analysis (PCA) revealed that the transcriptome of naïve m-reMAIT cells clustered upon adoptive transfer, which was similar to that of endogenous MAIT cells from lamina propria lymphocytes (*Figure 2—figure supplement 1*). We then focused on the transcripts relevant to the identity and function of MAIT cells (*Koay et al., 2019*; *Salou et al., 2019*). We found that the expression of transcripts, such as *Il7r*, *Il12rb*, *Il18r*, *Il18rap*, *Prf*, *Gzmb*, *Tnf*, *Ifng*, *Il22*, *Slamf7*, *Abcb1a* (encoding the efflux pump), *Zbtb16* (encoding PLZF), *Tbx21* (encoding T-bet), and *Zfp683* (encoding Hobbit), was upregulated upon the adoptive transfer of naïve m-reMAIT cells, and the expression levels of some transcripts reached those in endogenous MAIT cells (*Figure 2—figure supplement 2*). Similar results were obtained for transcripts pertinent to the tissue repairing and tissue residency signature (*Figure 2—figure supplements 3 and 4*; *Leng et al., 2019*; *Salou et al., 2019*; *Yanai et al., 2016*). Transcripts relevant to the V-D-J recombination machinery, such as *Rag1*, *Rag2*, *Hmgb1*, *Dntt*, and *Lig4*, were silenced across the organs upon adoptive transfer (*Figure 2—figure supplement 5*).

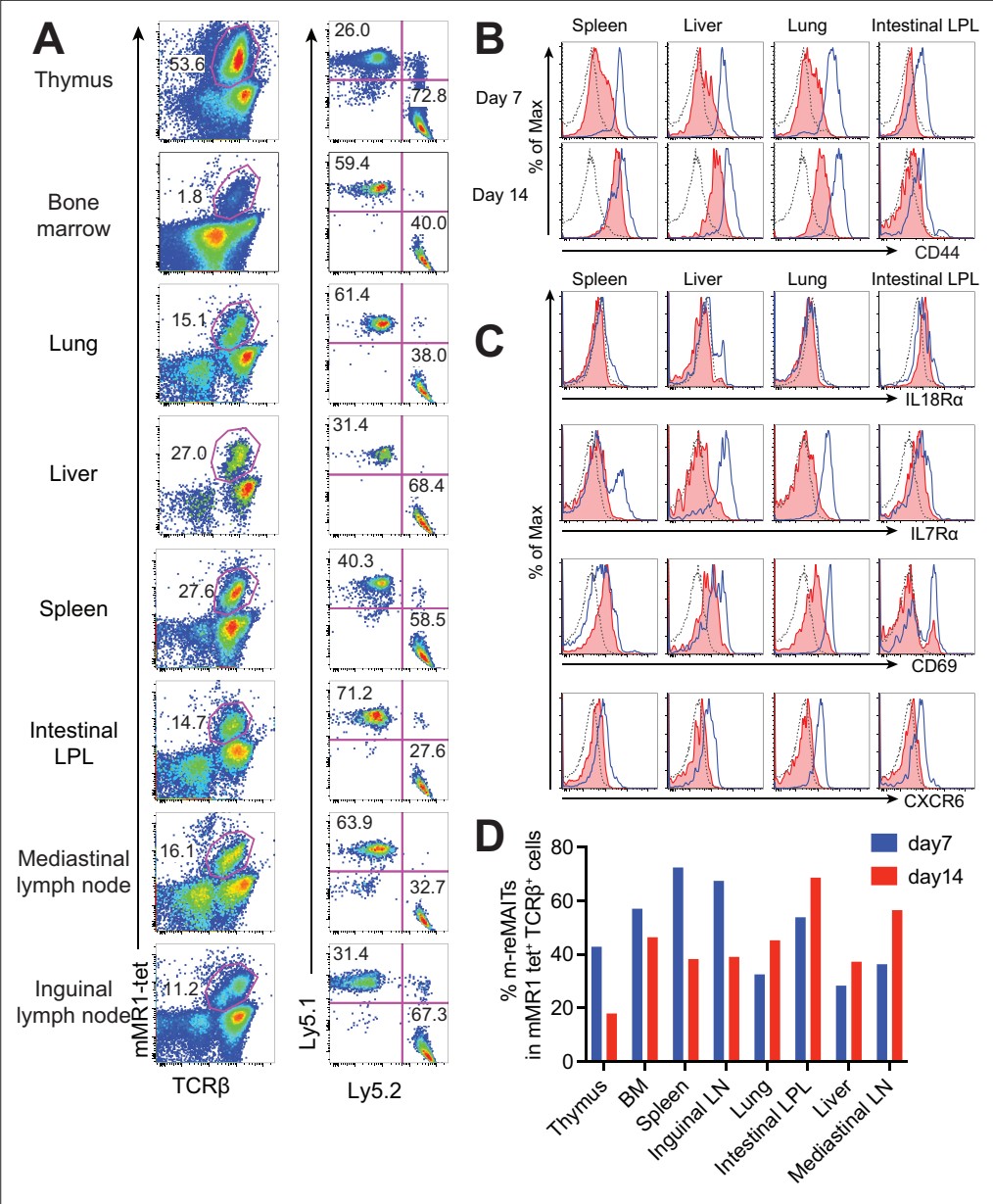

**Figure 2.** Behavior of m-reMAIT cells upon adoptive transfer. (**A**) m-reMAIT cell migration into different organs. m-reMAIT cells (Ly5.2) were adoptively transferred into C57BL/6 (Ly5.1) mice via an intraperitoneal injection (1 × 10⁷ cells/mouse), and endogenous as well as exogenous TCRβ⁺mMR1-tet⁺ cells were enriched 7 days later with mMR1-tet (left panel) and subjected to analyses of the expression of Ly5.1 and Ly5.2 (right panel). The number in the panel shows the percentage of Ly5.1 (endogenous) and Ly5.2 (exogenous) TCRβ⁺mMR1-tet⁺ cells. Representative data from a pool of 3–4 mice per experiment are shown. (**B**) Time-dependent upregulation of CD44. CD44 expression in TCRβ⁺mMR1-tet⁺ cells from the indicated tissues 7 and 14 days after m-reMAIT cell adoptive transfer. Isotype control (dotted line), endogenous mucosal-associated invariant T (MAIT) cells (Ly5.1⁺TCRβ⁺mMR1-tet⁺ cells) (plain blue line), and exogenous m-reMAIT cells (Ly5.2⁺TCRβ⁺mMR1-tet⁺ cells) (shaded in red). Representative data from a pool of 3–4 mice per experiment are shown. (**C**) Expression of molecules relevant to MAIT cells. The expression of molecules in m-reMAIT cells from the indicated tissues 14 days after adoptive transfer. Isotype control (dotted line), endogenous MAIT cells (Ly5.1⁺TCRβ⁺mMR1-tet⁺ cells) (plain blue line), and exogenous m-reMAIT cells (Ly5.2⁺TCRβ⁺mMR1-tet⁺ cells) (shaded in red). Representative data from a pool of 3–4 mice per experiment are shown. (**D**) Frequency of m-reMAIT cells. The frequency of m-reMAIT cells (m-reMAITs) among mMR1-tet⁺TCRβ⁺ cells harvested on days 7 and 14 from the indicated organs is shown. Representative data from two experiments are indicated.

*Figure 2 continued on next page*

*Figure 2 continued*

The online version of this article includes the following source data and figure supplement(s) for figure 2:

**Figure supplement 1.** Principal component analysis (PCA).

**Figure supplement 2.** Representative transcripts relevant to mucosal-associated invariant T (MAIT) cell identity and function.

**Figure supplement 2—source data 1.** Expression of the genes relevant to mucosal-associated invariant T (MAIT) cell identity and function.

**Figure supplement 3.** Transcripts relevant to tissue repairing.

**Figure supplement 3—source data 1.** Expression of the genes relevant to tissue repairing.

**Figure supplement 4.** Transcripts relevant to tissue residency.

**Figure supplement 4—source data 1.** Expression of the genes relevant to tissue residency.

**Figure supplement 5.** Transcripts relevant to V-D-J recombination.

**Figure supplement 5—source data 1.** Expression of the genes relevant to V-D-J recombination.

These results indicated that m-reMAIT cells migrated into different organs accompanying maturation upon adoptive transfer in immunocompetent mice.

## Tumor inhibitory activity of m-reMAIT cells

Since the role of MAIT cells in tumors has been a target for intensive scrutiny, we attempted to clarify whether m-reMAIT cells interfered with tumor metastasis and prolonged mouse survival (*Godfrey et al., 2015*; *Toubal et al., 2019*). A previous study on B16F10 melanoma suggested that MAIT cells induce tumor development in a manner that is dependent on MR1 and agonists, such as 5-OP-RU (*Yan et al., 2020*). While B16F10 significantly upregulated MR1 on the cell surface with the 5-OP-RU challenge, a similar change was not observed for Lewis lung carcinoma (LLC) (*Figure 3A*). Therefore, the use of LLC allows for assessments of the role of m-reMAIT cells in tumor immunosurveillance independent of the 5-OP-RU-MR1 axis.

The adoptive transfer of m-reMAIT cells into mice followed by the LLC inoculation prolonged survival in a dose-dependent manner. While $3 \times 10^5$ m-reMAIT cells slightly prolonged survival over the control, more than $1 \times 10^6$ m-reMAIT cells significantly prolonged survival (*Figure 3B*). Nevertheless, multiple transfers of m-reMAIT cells failed to significantly prolong survival (*Figure 3C*). In contrast to these results, the adoptive transfer of m-reMAIT cells did not suppress tumor growth at any dose following the subcutaneous inoculation of LLC, which represents an in situ tumor growth model (*Figure 3D*), irrespective of the delayed emergence of m-reMAIT cells in the skin (*Figure 3—figure supplement 1*).

These results implied that m-reMAIT cells inhibited tumor metastasis rather than suppressing tumor growth in situ.

## Cytolytic activity of m-reMAIT cells in combination with NK cells

While the above results indicated that m-reMAIT cells functioned to suppress metastasis, it currently remains unclear whether other immune cells are involved in this tumor inhibitory activity. NK cells are an essential innate sentinel in tumor immunosurveillance, and previous studies reported contradicting findings regarding the role of MAIT cells on NK cells in tumor immunity. Yan et al. demonstrated that MAIT cells promoted tumor growth by suppressing the activity of NK cells and T cells (*Yan et al., 2020*), while others suggested that the activation of MAIT cells in vivo strengthened antitumor activity concomitant with an enhanced NK cell response (*Petley et al., 2021*). Therefore, we investigated whether and how m-reMAIT cells and NK cells mutually affect their functions in tumor immunity. Interactions between NK cells and m-reMAIT cells were assessed in a coculture. While m-reMAIT cells were activated in an NK cell dose-dependent manner, the activation of NK cells was less prominent (*Figure 4A*). Moreover, the coculture led to the production of Th1 and Th17 cytokines as well as inflammatory chemokines in an NK cell dose-dependent manner (*Figure 4B*).

To obtain further insights into the molecular events underlying this interaction, NK cells and m-reMAIT cells were purified with FACS after the coculture and each subset was examined for transcripts relevant to cytolytic function (*Figure 4C and D*). *Ifng* was induced in m-reMAIT cells, but not in NK

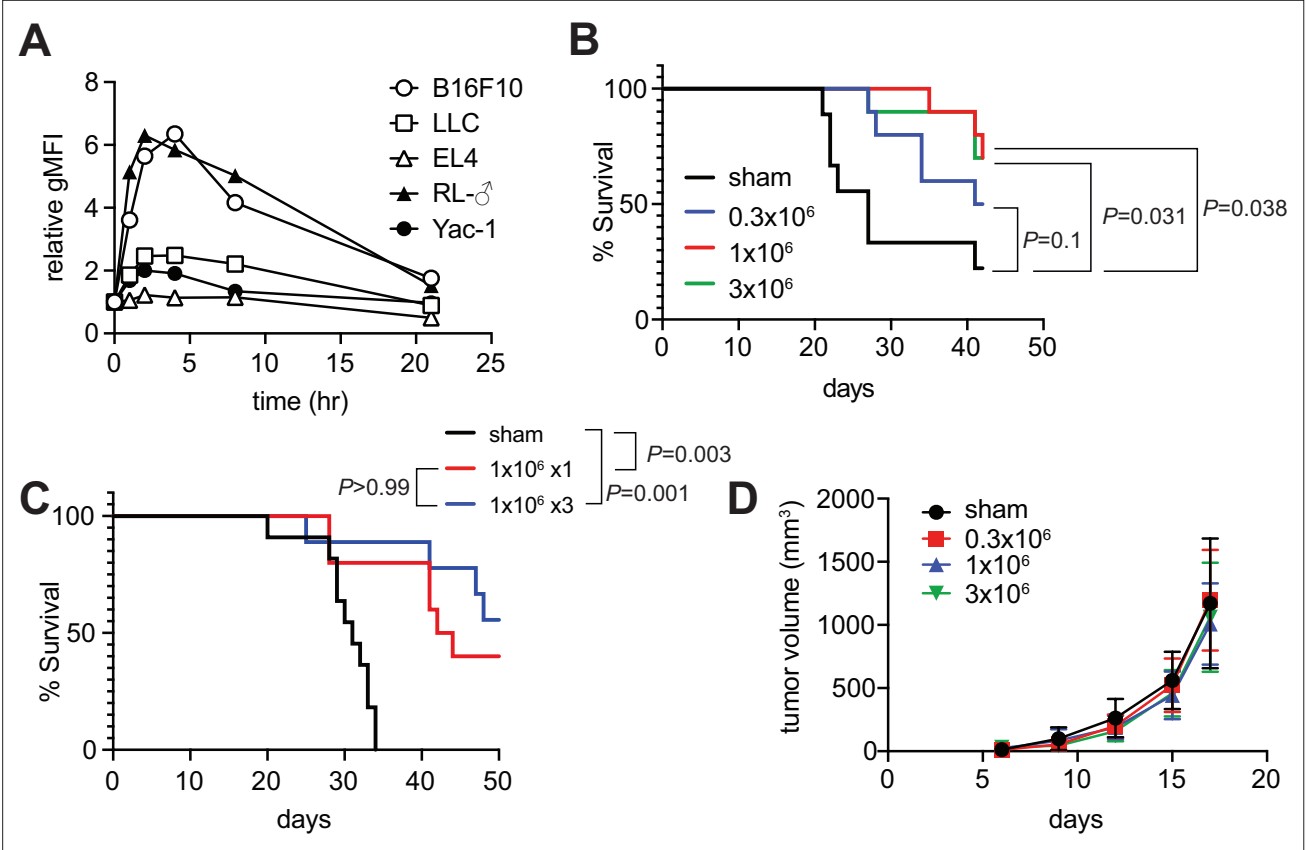

**Figure 3.** Tumor inhibitory activity of m-reMAIT cells. (**A**) Time course of 5-(2-oxopropylideneamino)-6-D-ribitylaminouracil (5-OP-RU)-dependent MR1 expression. The indicated cancer cell lines were challenged with 5-OP-RU. MR1 expression levels on the cell surface at the indicated time point are shown as relative geometric mean fluorescent intensity (gMFI). Data are representative of three independent experiments. (**B**) m-reMAIT cell dose-dependent survival extension. C57BL/6 mice received the indicated amounts of m-reMAIT cells 6 days prior to the Lewis lung carcinoma (LLC) inoculation ($3 \times 10^5$ cells/mouse i.v.), and survival was monitored (n = 10–12/group). Data are representative of three independent experiments. p-Values between the indicated group are shown (the log-rank test). (**C**) Effects of the multiple transfers of m-reMAIT cells on survival. The survival of C57BL/6 mice that received m-reMAIT cells ($1 \times 10^6$/mouse, i.p.) 6 days prior to the LLC inoculation ($3 \times 10^5$ cells/mouse, i.v.), and of mice that received LLC and two more consecutive transfers of m-reMAIT cells ($1 \times 10^6$/transfer/mouse) was monitored (n = 10–12/group). Sham-treated mice that only received LLC served as a control. Data are representative of two independent experiments. p-Values between the indicated groups are shown (the log-rank test). (**D**) Effects of m-reMAIT cells on in situ tumor growth. Growth curve of LLC. LLC ($3 \times 10^5$ /mouse) was subcutaneously inoculated into the right flank of C57BL/6 mice 6 days after the m-reMAIT cell transfer (i.p.). Tumor size was plotted with time. Sham treated (●), $0.3 \times 10^6$ transferred (■), $1.0 \times 10^6$ transferred (▲), and $3.0 \times 10^6$ m-reMAIT cells transferred (▼). Data are shown as SEM (5–6 mice per group).

The online version of this article includes the following source data and figure supplement(s) for figure 3:

**Source data 1.** Time-dependent MR1 expression in various cancer cell lines upon 5-(2-oxopropylideneamino)-6-D-ribitylaminouracil (5-OP-RU) challenge.

**Figure supplement 1.** m-reMAIT cells in the skin.

**Figure supplement 1—source data 1.** Delayed emergence of m-reMAIT cells in the skin upon adoptive transfer.

cells. Similarly, the coculture enhanced the transcripts for serine proteases, such as *Grza*, *Grzb*, and *Tnf*, in m-reMAIT cells and NK cells (***Trapani and Smyth, 2002***). In contrast, the expression of the transcripts for the TNF superfamily, including *Fasl* and *Tnsf10*, was only stimulated in m-reMAIT cells. Moreover, the expression of *Ccl4 (Mip1b)*, *Ccl5 (Rantes)*, *Il17a*, and *Il6* was upregulated in m-reMAIT cells in contrast to only *Ccl5 (Rantes)* in NK cells.

We then investigated whether this change had an impact on CD69 and CD107a, a marker of cytolytic granule exocytosis. While the coculture with m-reMAIT cells did not markedly upregulate the expression of CD69 (CD69$^+$ cells) in NK cells, the addition of Yac-1, an NK cell-sensitive tumor cell line (alone or in combination with m-reMAIT cells), increased the percentage of CD69$^+$ cells. However, these combinations resulted in negligible changes in the percentage of CD69$^+$CD107a$^+$ cells

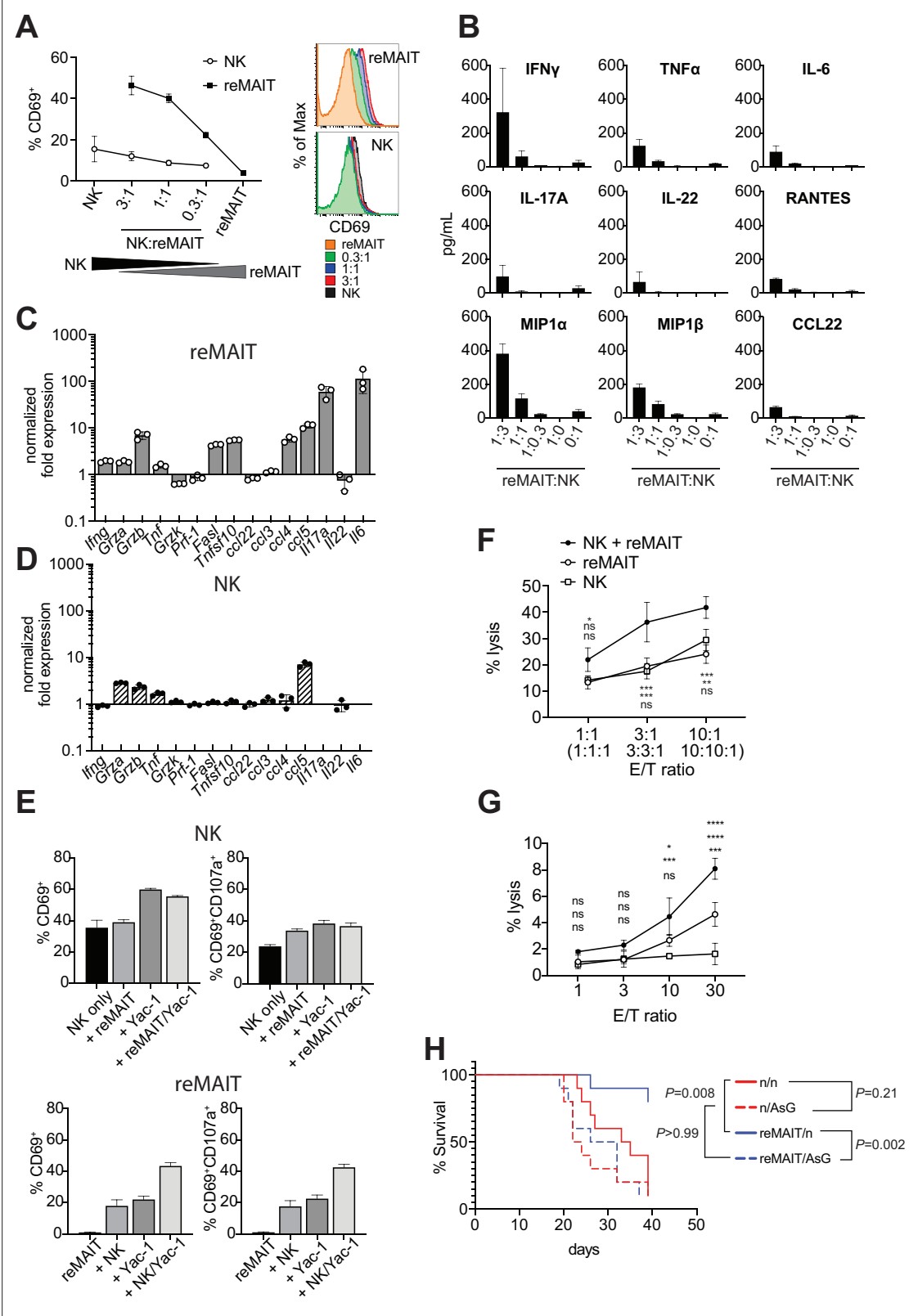

**Figure 4.** Antitumor activity of m-reMAIT cells bolstered by NK cells. (**A**) Activation of m-reMAIT cells by NK cells. CD69 expression in m-reMAIT cells (reMAIT ●) and NK cells (NK ○) upon incubation at the indicated ratio. The percentage of cells expressing CD69 (left panel) and the intensity of CD69 in each cell population (right panel) are shown. (**B**) Cytokines and chemokines upon a coculture. Cytokines and chemokines released upon a coculture of m-reMAIT cells and NK cells at the indicated ratio are shown (reMAIT:NK). Amounts were quantified with LegendPlex. Representative data from two

*Figure 4 continued on next page*

*Figure 4 continued*

independent experiments are shown. (**C**) Transcripts relevant to cytolytic activity in m-reMAIT cells. *Ifng, Gzma, Gzmb, Tbf, Gzmk, Pfr1, Fasl, Tnfsf10, Il6, Il17a, Il22, Ccl3, Ccl4, Ccl5,* and *Ccl22* in m-reMAIT cells cultured with NK cells were quantified with qRT-PCR. m-reMAIT cells and NK cells were sort-purified after the coculture (purity >98%) or cultured individually. The expression of each transcript was normalized with *Gapdh,* and fold changes in the relative expression of the transcript in m-reMAIT cells cultured with NK cells relative to that in m-reMAIT cells cultured alone are shown. Data are representative of three independent experiments. (**D**) Transcripts relevant to cytolytic activity in NK cells. Fold changes in the relative expression of the indicated transcript as described in (**C**) in NK cells cultured with m-reMAIT cells relative to that in NK cells alone are shown. Representative data from three independent experiments are shown. (**E**) Activation and degranulation of NK cells and m-reMAIT cells. The expression of CD69, an activation marker, and CD107a, a marker for the exocytosis of cytolytic granules, was assessed under various culture conditions. The percentages of CD69$^+$ cells and CD69$^+$CD107a$^+$ cells among NK cells alone (control), NK cells cocultured with m-reMAIT cells (+reMAIT), NK cells cultured with Yac-1 (+Yac-1), and NK cells cocultured with m-reMAIT cells and Yac-1 (+reMAIT/Yac-1) (upper panels). The percentages of CD69$^+$ cells and CD69$^+$CD107a$^+$ cells among m-reMAIT cells alone (control), m-reMAIT cells cocultured with NK cells (+NK), m-reMAIT cells cocultured with Yac-1 (+Yac-1), and m-reMAIT cells cocultured with NK cells and Yac-1 (+NK/Yac-1) (lower panels). Data are representative of three independent experiments. (**F**) Cytolytic activity against Yac-1. Cytolytic activity of m-reMAIT cells (reMAIT ○), NK cells (NK □), and NK cells plus m-reMAIT cells (NK+reMAIT ●). Cytolytic activities (% lysis) at different effector (NK cells, m-reMAIT cells, and NK cell+m-reMAIT cells)/Target (Yac-1) (E/T) ratios are shown. Representative data from three experiments are shown. The significance of differences between the groups at the indicated E/T ratio assessed with a two-way ANOVA is shown (*p<0.05, **p<0.01, ***p<0.005). From the top, NK+reMAIT vs. reMAIT, NK+reMAIT vs. NK, and reMAIT vs. NK. Data are representative of three independent experiments. (**G**) Cytolytic activity against Lewis lung carcinoma (LLC). The cytolytic activities of m-reMAIT cells (○), NK cells (□), and m-reMAIT cells plus NK cells (●) against LLC at the indicated E/T ratio are shown as % lysis. The significance of differences between the groups is calculated as in (**E**). Data are representative of three independent experiments. (**H**) NK cell-dependent extension of survival. C57BL/6 mice were divided into two groups, one that received 1 × 10$^6$ m-reMAIT cells (reMAIT) and another that was left untreated (n). Each group was further divided into two subgroups, one that received consecutive injections of anti-Asialo GM1 (AsG) (–1 and 16 days, 50 μg/mouse) after the LLC inoculation (3 × 10$^5$ i.v.) and another that was left untreated (n). Survival was monitored thereafter. Representative data from two independent experiments are shown (n = 10–14/group). p-Values between the indicated groups are shown (the log-rank test).

The online version of this article includes the following source data for figure 4:

**Source data 1.** Antitumor activity of m-reMAIT cells bolstered by NK cells.

(*Figure 4E*, upper panels). In contrast, incubation with NK cells or Yac-1 enhanced the percentage of CD69$^+$ cells in m-reMAIT cells. Moreover, a coculture with NK cells together with Yac-1 resulted in further increases, and similar results were obtained for CD69$^+$CD107a$^+$ cells in m-reMAIT cells (*Figure 4E*, lower panels).

We then examined whether the above activation and/or expression of CD107a reflected lytic activity against tumor cells. m-reMAIT cells lysed Yac-1, an NK cell-sensitive tumor cell line, as efficaciously as NK cells, and these cells synergistically enhanced killing (*Figure 4F*). We extended our study to LLC, an NK cell-insensitive cell line. NK cells alone did not induce cell death in LLC, whereas m-reMAIT cells did. Moreover, the combination of these cells exhibited synergistically enhanced lytic activity (*Figure 4G*).

We then investigated the significance of the interaction between NK cells and m-reMAITs cell in vivo. To assess the role of NK cells in m-reMAIT cell-mediated anti-metastasis activity, NK cells were depleted in mice with an anti-Asialo GM1 antibody (AsG). This depletion did not significantly affect the survival of control mice. However, the AsG treatment abrogated the survival advantage of m-reMAIT cells, highlighting the stimulatory role of NK cells in m-reMAIT cell-mediated tumor inhibitory activity (*Figure 4H*).

These results demonstrated that the intrinsic cytolytic activity of m-reMAIT cells against tumors was bolstered by NK cells concomitant with the upregulation of the relevant arsenals.

## Discussion

The present results revealed that m-reMAIT cells were competent in TCR signaling. The activation of m-reMAIT cells and subsequent production of cytokines and chemokines in the absence of APCs suggested that 5-OP-RU directly operated via MR1 in m-reMAIT cells (*Figure 1D–F*). 5-OP-RU may be passively transported into cells, which in turn promotes the release of antigen-loaded MR1 from the endoplasmic reticulum (ER) onto the plasma membrane, thereby enabling signaling via TCR in m-reMAIT cells (referred to as a *cis*-acting signal) regardless of the apparent absence of MR1 on the cell surface (*Figure 1B and D*, *Figure 1—figure supplement 1I*; *McWilliam et al., 2016*). In contrast, mMR1-tet directly binds to TCR on m-reMAIT cells and elicits signals (referred to as *trans*-acting

signals). Therefore, 5-OP-RU and mMR1-tet may have elicited similar signals through serine/threonine and/or tyrosine phosphorylation or dephosphorylation via TCR, which ultimately resulted in a different profile of cytokine and chemokine production (*Figure 1F–H and J*; *Gaud et al., 2018*; *Nausch and Cerwenka, 2008*). In this respect, the signaling pathway(s) responsible for the production of IL-2, IL-4, MIP1α CCL3 (MIP-1α), and CCL4 (MIP-1β) may be common between 5-OP-RU and mMR1-tet. In contrast, mMR1-tet may have activated the signaling pathways pertinent to the activation of Th17 as evidenced by the production of Th17 cytokines, such as IL-17F, IL-22, and IL-23, concomitant with inflammatory chemokines, including CCL5, CCL2, CXCL1, and CXCL6, in addition to TNF-α (*Figure 1F*). These results suggest that mMR1-tet signals, in part, through NF-κB, while 5-OP-RU does not (*Liu et al., 2017*).

While 5-OP-RU induced protein tyrosine phosphorylation, the identity of these proteins remains unclear, except for LAT (*Figure 1G and H*). Since LAT forms a hub that interacts with many signaling molecules, such as transmembrane receptors, phosphatases, kinases, guanine nucleotide exchange factors, GTP-activating proteins, ion channels, transporters, and ubiquitin ligases, further studies will provide insights into the proteins responsible for MAIT-TCR signaling and reveal the differences in signaling between conventional T cells and MAIT cells (*Malissen et al., 2014*).

It is important to note that the challenge with mMR1-tet induced the activation of m-reMAIT cells and production of cytokines and chemokines. This suggests that MAIT cells prepared from animals, including humans and mice, are unintentionally stimulated during experiments. Therefore, the interpretation of data requires caution (*Figure 1E, F and J*).

Although a previous study suggested that MAIT cells comprise MAIT1 and MAIT17 subsets characterized by the Th1 and Th17 transcriptomes, respectively, the present results revealed that m-reMAIT cells produced Th1, Th2, and Th17 cytokines (*Salou et al., 2019*). This may reflect a developmental stage at which m-reMAIT cells are en route to functional maturation and m-reMAIT cells generated in vitro represent the most immature stage. Therefore, it is tempting to postulate that the interaction between nascent MAIT cells and double-positive thymocytes favors the differentiation of naïve MAIT cells into MAIT1 and MAIT17 in vivo. The results on the adoptive transfer of m-reMAIT cells indicated that nascent MAIT cells egressed from the thymus migrate into any organs concomitant with the upregulation of CD44, a cell adhesion receptor, memory, and/or activation marker (*Figure 2B*; *Benlagha et al., 2005*). It is of interest that the expression of IL-7Rα in m-reMAIT cells became quasi-equivalent to that in endogenous cells in the intestines. Since IL-7 from intestinal epithelial cells plays a pivotal role in their homeostasis, the upregulation of IL-7Rα in m-reMAIT cells may mirror an intrinsic and primary role in this homeostasis and/or in the integrity of the intestinal epithelial barrier (*Rouxel et al., 2017*; *Shalapour et al., 2012*). Alternatively, it may reflect the role of MAIT cells in IL-17A and Th1 cytokine production, which is dependent on IL-7 (*Tang et al., 2013*). Taken together with the time-dependent upregulation of CD44, increases in IL-18Rα and CXCR6 across organs indicated that m-reMAIT cells matured in the host. Thus, naïve m-reMAIT cells, defined as mMR1-tet$^+$TCRβ$^+$ cells generated in vitro, may be promoted maturation in terms of transcriptional program upon adoptive transfer into the recipient mice, as evidenced by the acquisition of cytokine receptors, the effector molecules involved in cytotoxicity and tissue repairing, and those relevant to tissue residency (*Figure 2—figure supplements 2–4*). Moreover, the silencing of transcripts relevant to V-D-J recombination may ensure the continuum of the identity of m-reMAIT cells in the recipient (*Figure 2—figure supplement 5*). These results have important implications for future cell therapy with human iPSC-derived MAIT cells. Further studies are needed to clarify whether and how the adoptive transfer of naïve m-reMAIT cells affects epigenetic states, and whether this change leads to the generation of bona fide MAIT cells.

m-reMAIT cell dose-dependent mouse survival indicated their tumor inhibitory activity and function as effector cells in tumor surveillance (*Figure 3B*). However, since three consecutive injections of m-reMAIT cells failed to confer survival superiority, the niche for MAIT cells appeared to be saturated. Therefore, further studies are warranted to investigate whether a lymphopenia-inducing regimen, such as irradiation and/or cyclophosphamide, confers a difference in survival.

The failure of m-reMAIT cells to inhibit tumor growth in situ regardless of their mobilization to the skin indicated their poor infiltration into tumors rather than the loss of antitumor activity (*Figure 3—figure supplement 1*). Therefore, future research that focuses on whether an intratumor injection of m-reMAIT cells inhibits tumor growth is needed.

While the importance of NK cells in tumor surveillance is well known, that of MAIT cells remains elusive (*Cogswell et al., 2021*; *Guillerey et al., 2016*; *Sharma et al., 2017*). The present results showing the NK cell-dependent activation of m-reMAIT cells and the subsequent production of inflammatory cytokines and chemokines suggested that NK cells stimulated the tumor inhibitory effects of MAIT cells (*Figure 4A and B*). Moreover, the enhanced expression of transcripts relevant to the cytolytic function of m-reMAIT cells upon an interaction with NK cells in vitro indicated that m-reMAIT cells exhibited cytolytic activity in vivo (*Figure 4C–G*). The upregulation of *Grzb*, *Fasl*, and *Tnfsf10* in m-reMAIT cells suggested that cytolytic activity comprised exocytosis- as well as caspase-mediated killing (*Rossin et al., 2019*; *Trapani and Smyth, 2002*). NK cell depletion compromised the survival of mice, which is consistent with the above findings, further highlighting the role of NK cells in boosting the tumor inhibitory activity of m-reMAIT cells in vivo. While NK cells recognize NKG2D ligands and/or the lack of MHC I on tumor cells, the molecular mechanisms by which m-reMAIT cells recognize and eliminate tumor cells warrant further study (*Nausch and Cerwenka, 2008*).

Contrary to our results, MAIT cells have been shown to induce the metastasis of melanoma B16F10 (*Yan et al., 2020*). Although the reason for this discrepancy currently remains unclear, differences in the dynamics of MR1 shuttling between ER and the plasma membrane upon a 5-OP-RU challenge or putative ligand(s) present in the tumor milieu (TM) may be a key feature. Since LLC did not strongly upregulate MR1 on the cell surface when stimulated (*Figure 4A*), 5-OP-RU may have bolstered B16F10 tumorigenicity through MR1. The MR1-elicited signal may suppress the function of NK cells for B16F10, whereas that from putative ligand(s) present in TM via MR1 did not, thereby enhancing or preserving NK cell functions. Further studies are needed to elucidate the underlying mechanisms, which will provide a novel avenue for tumor immunotherapy with iPSC-derived MAIT cells.

Although we showed that m-reMAIT cells exhibited cytolytic activity against LLC together with NK cells in vitro (*Figure 4C and E–G*), difficulties are associated with demonstrating that m-reMAIT cells isolated from mice exhibit similar lytic activity ex vivo. This is due to the low recovery of m-reMAIT cells and the compulsory use of mMR1-tet in cell preparations, which may interfere with cytolytic activity. Furthermore, it currently remains unclear whether these results are applicable to other cancer cells, such as B16F10, if the use of 5-OP-RU enhances or inhibits the cytolytic activity of m-reMAIT cells. Furthermore, since the transcriptome in m-reMAIT cells upon adoptive transfer still differed from that in endogenous cells, caution is needed when extrapolating the present results to MAIT cells in vivo.

In summary, the adoptive transfer model with m-reMAIT cells used in this study opens a new avenue for exploiting the function of MAIT cells and providing insights into their interaction(s) with immune cells in immunity.

## Materials and methods

### Mice

All mouse experiments were performed with approval from the Institutional Animal Care and Use Committee of Dokkyo Medical University (permit number 1215). C57BL/6NJcl mice were purchased from CLEA Japan (Tokyo, Japan). C57BL/6 (Ly5.1) mice were obtained from the RIKEN Bioresource Center and bred in-house. All mice were housed in the Animal Research Center, Dokkyo Medical University, under specific pathogen-free conditions with controlled lighting and temperature with food and water provided ad libitum. Male C57BL/6NJcl mice aged 6 weeks were used to isolate MAIT cells for the generation of iPSCs. Male and female mice aged between 8 and 12 weeks were used in adoptive transfer experiments and tumor experiments.

### Cell lines

The cell lines used in this study were obtained as described in the Key resources table. These cells were periodically checked for mycoplasma contamination with mycoplasma detection kit (Takara Bio, Japan). Mycoplasma was not detected throughout the experiments. OP9/DLL1 cells were maintained in αMEM supplemented with 20% FBS. The mouse cancer cell lines B16F10, CH27, CH27/mMR1, EL4, and LLC were cultured in DMEM supplemented with 10% FBS, while RL-♂1, WT3, WT3/mMR1, and Yac-1 were cultured in RPMI 1640 supplemented with 10% FBS at 37°C in 5% $CO_2$.

## Antibodies

The antibodies used in this study are listed in the Key resources table.

## Oligonucleotides

The oligonucleotides used in this study are summarized in the Key resources table.

## Preparation of mouse immune cells

### Spleen, thymus, and lymph nodes

Tissues were prepared by mashing through a 40 μm mesh cell strainer with a syringe plunger. Single cells were suspended in RPMI 1640 supplemented with 10% FBS, 10 mM HEPES pH 7.0, 0.1 mM 2-mercaptoethanol, and 100 IU/ml of penicillin/streptomycin (referred to as cR10) and spun down at 400 × $g$ for 4 min. To lyse red blood cells, the cell pellet was suspended in autoclaved ice-cold MilliQ water for 15 s and immediately neutralized with 4% FBS in 2× PBS. After centrifugation, cells were resuspended in cR10.

### Lungs and liver

Single-cell suspensions from the lungs and liver were prepared using enzymatic digestion. Briefly, tissues were placed into a GentleMACS C-tube (Miltenyi Biotec) and cut into approximately 5 mm³ pieces. 4 ml of tissue digestion solution (90 U/ml collagenase Yakult, 275 U/ml collagenase type II, 145 PU/ml Dispase II, and 4% BSA in HBSS) was added per tissue, and tissues were homogenized using the GentleMACS dissociator (Miltenyi Biotec) with the following program: m_lung_01_02 for the lungs and m_liver_03_01 for the liver. Suspensions were then incubated at 37°C for 30 min under gentle rotation, followed by dissociation with m_lung_02_01 for the lungs and m_liver_04_01 for the liver, and subjected to discontinuous density centrifugation over layers of 40 and 60% Percoll at 400 × $g$ for 20 min. Cells were recovered from the 40–60% Percoll interface, washed with PBS, and then suspended in cR10.

### Intestines

The intestines were longitudinally incised, and their contents were thoroughly washed out three times by vigorous shaking. Tissues dissected into 1 cm pieces were placed into a 50 ml conical tube and washed vigorously by shaking three times with PBS. After discarding the supernatant, the tissues were treated with 40 ml of intraepithelial lymphocyte-washing solution (HBSS containing 1 mM DTT, 5 mM EDTA, and 1% BSA) by shaking vigorously at 37°C for 30 min under gentle rotation. After washing three times with MACS buffer (PBS containing 2 mM EDTA and 0.5% BSA), the tissues were washed again with 40 ml HBSS. After removal of the supernatant, the tissues were placed into a GentleMACS C-tube (Miltenyi Biotec), cut into small pieces with scissors, and 4 ml of the tissue digestion solution (see above) was added. The tissues were processed using the program m_brain_01_02 and digested at 37°C for 30 min under gentle rotation followed by dissociation with the program m_intestine_01_01. Cell suspensions were subjected to Percoll discontinuous density centrifugation, and isolated cells were resuspended in cR10.

### Skin

Skin samples were prepared on days 6 and 9 after adoptive transfer of m-reMAIT cells (intraperitoneally [i.p.] 1.0 × 10⁷ cells/mouse) from the right flank (20 mm × 20 mm) without LLC or on day 9 with LLC (LLC was subcutaneously injected at the right flank 3 days post m-reMAIT cell transfer). Skins were incised and digested with HBSS containing collagenase II (1.2 mg/ml, Worthington), Yakult collagenase (0.8 mg/ml, Yakult), thermolysin (0.5 mg/ml, Sigma), DNase I (0.1 mg/ml, Roche), and BSA (40 mg/ml) at 37°C for 2 hr under gentle shaking, followed by filtration through the cell strainer (100 μm). MAIT cells and m-reMAIT cells were then enriched with APC-labeled mMR1-tet (NIH tetramer core facility) followed by anti-APC-Microbeads (Miltenyi Biotec), and stained with anti-CD45, -Ly5.1, -Ly5.2, and -TCRβ antibodies. The number of m-reMAIT cells (CD45⁺ Ly5.2⁺ TCRβ⁺ mMR1-tet⁺) was measured with flow cytometry.

### Generation of iPSCs from mouse MAIT cells

Single-cell suspensions from the lungs were stained with APC-labeled 5-OP-RU-loaded mouse MR1 tetramer (mMR1-tet) (NIH Tetramer Core Facility) before magnetic enrichment with anti-APC microbeads and LS or MS columns (Miltenyi Biotec) according to the manufacturer's instructions. Enriched cells were stained with FITC-CD44, PE-B220, PE-F4/80, and PE/Cy7-TCRβ and MAIT cells were then sorted as B220⁻F4/80⁻TCRβ⁺CD44^hi mMR1-tet⁺ cells with the FACSJazz cell sorter (BD Biosciences). Regarding transduction, purified MAIT cells (1800 – 6300 cells) were placed into the wells of 96-well plastic plates coated with anti-CD3 and anti-CD28 antibodies (15 and 20 µg/ml, respectively) for 18 hr and then infected with Sendai virus KOSM302L at MOI of 25 at 37°C in 5% $CO_2$ under gentle orbital shaking for 2.75 hr. After transduction, the virus-containing medium was removed by centrifugation at 400 × *g* for 4 min. Cells were refed with mouse ES medium containing StemSure D-MEM (FUJIFILM Wako), 15% FBS (BioSera), 1× nonessential amino acids (FUJIFILM Wako), 2 mM L-glutamine (Nacalai Tesque), 0.1 mM 2-mercaptoethanol (FUJIFILM Wako), 100 U penicillin/100 µg streptomycin (Lonza), 1000 U/ml mouse LIF (FUJIFILM Wako), and transferred to the wells of 6-well culture plates seeded with mitomycin C (MMC)-treated MEF. Medium was changed every 2–3 days. 3–4 weeks after infection, ES-like colonies were picked up and treated with 0.25% Trypsin-1 mM EDTA (TE) at 37°C for 10 min and individually transferred to the wells of 24-well culture dishes seeded with MMC-treated MEF. Among the 46 iPSCs isolated after reprogramming, 5 iPSCs (L3, L7, L11, L15, and L19) were subjected to limiting dilutions after 2–3 passages and a monoclonal iPSC line was generated and designated accordingly (e.g., L3-1, L7-1 to -8, L11-1 to -4, L15-1 to -3, and L19-1 to -6) (*Figure 1—figure supplement 1B and D*). The established iPSC lines were maintained with mouse ES medium supplemented with 3 µM CHIR99021 (FUJIFILM Wako) and 10 µM PD0325901 (FUJIFILM Wako). The majority of experiments were performed with L7-1, except for the PCR analysis in *Figure 1—figure supplement 1A*, in which the samples were used before limiting dilutions. Monoclonal iPSC lines were cryopreserved with serum-free cell culture freezing medium BAMBANKER (GC lymphotec) in liquid nitrogen.

### PCR detection of the rearranged configuration of *TCR* loci in MAIT-iPSCs

To confirm that MAIT-iPSCs stemmed from MAIT cells, PCR detecting the rearranged configuration of *Trav* specific for MAIT cells was performed with the primer sets ADV19 and AJ33. Genomic DNA was prepared from MAIT-iPSCs with NaOH. To detect *Trbv* in MAIT-iPSCs, total RNA was prepared from m-reMAIT cells (days 20–28 of differentiation, varying according to the clones) using the RNeasy Mini kit (QIAGEN). cDNA was synthesized with the first-strand cDNA synthesis kit (Thermo Fisher Scientific) and subjected to PCR with the primer sets TRBV13 and TRBC-Rev, and TRBV19 and TRBC-Rev followed by DNA sequencing (Fasmac). The usage of TRBV-D-J was analyzed using IGBLAST (NCBI, NIH https://www.ncbi.nlm.nih.gov/igblast/).

### Southern blot analysis to detect rearranged *Trav*

Genomic DNA (3 µg) prepared from MAIT-iPSCs (L7–1 to -6, L11-1 to -4, L15-1 to -3, and L19-1 to -6) with NucleoSpin Tissue (MACHEREY-NAGEL) was digested with 40 U *BamH*1 (TOYOBO) at 37°C for 18 hr, separated by 0.9% agarose gel electrophoresis, and transferred onto Biodyne membranes (PALL Life Sciences). The membrane was pre-hybridized with PerfectHyb (TOYOBO) at 68°C for 60 min and hybridized with an α-³²P-dCTP-labeled probe at 68°C for 18 hr. The probe was amplified with the primer set TRAV19 (forward) and TRAV19 (reverse) using genomic DNA from C57BL/6 as a template, and the resultant 350 bp PCR product was radiolabeled with the Random Primer DNA Labeling Kit version 2.0 (Takara) (*Figure 1—figure supplement 1D*). The membrane was washed with 2× saline sodium citrate (SSC) and then 0.1× SSC buffer twice at 68°C, exposed to an imaging plate, and the signal was detected with a phosphoimager (Typhoon 7000, GE Healthcare).

### Differentiation of MAIT-iPSCs into MR1 tetramer⁺ m-reMAIT cells

On day 0, 1.2 × 10⁵ MAIT-iPSCs (L3-1, L7-1(L7), L11-1, L15-1, and L19-1) were seeded on a 10 cm culture dish containing 2–4 days post-confluent OP9/DLL1 in αMEM supplemented with 10% FBS (Corning). On day 3, culture medium was replaced with fresh medium. On day 5, cells were treated with 2 ml TE and 8 ml of αMEM containing 20% FBS (BioSera) was added and then suspended

in wells. The dish was incubated at 37°C in 5% $CO_2$ for 45 min. Floating cells were collected and passed through a 40 µm mesh cell strainer. After centrifugation, the cells were transferred to a new 10 cm culture dish filled with confluent OP9/DLL1 in αMEM containing 10% FBS (Corning) and 5 ng/ml human recombinant FLT3L. On day 8, loosely attached cells (cells containing lymphocyte progenitors) were harvested with gentle pipetting. After centrifugation, the cells were transferred to the wells of a 6-well culture plate containing confluent OP9/DLL1 in αMEM supplemented with 20% FBS (BioSera), 5 ng/ml human FLT3L, and 1 ng/ml mouse IL-7. On day 10 and later, culture medium was changed every other day (20% FBS containing αMEM supplemented with 5 ng/ml human FLT3L and 1 ng/ml mouse IL-7). Differentiated cells from iPSCs (m-reMAIT cells) were expanded on a new 10 cm culture dish filled with confluent OP9-DLL1 in αMEM supplemented with 20% FBS (BioSera) and mouse IL-7. m-reMAIT cells harvested on days 21–28 were used in the experiments unless otherwise indicated.

## Preparation of 5-OP-RU
An aliquot of 5-A-RU (3.6 mM in DMSO) was incubated with three volumes of 1 mM methylglyoxal (MilliQ water) at 37°C for 30 min and used for assays. The appropriate synthesis of 5-OP-RU was confirmed with LC-MS/MS (Triple Quad 5500, Sciex).

## m-reMAIT cell activation assay
m-reMAIT cells (L3-1, L7-1(L7), L11-1, L15-1, and L19-1) were incubated with the indicated concentration of 5-OP-RU or mMR1-tet at 37°C in 5% $CO_2$ for 18 hr in the presence or absence of equal numbers of CH27, CH27/mMR1, WT3, and WT3/mMR1, and then subjected to a CD69 expression analysis by flow cytometry. The culture supernatant was used for cytokine and chemokine quantification (see below). To examine whether the above activation of m-reMAIT cells was dependent on MR1, an anti-MR1 antibody (26.5) or isotype antibody (10 µg/ml) was added 1 hr prior to the addition of 5-OP-RU.

## Doubling time estimation with CFSD-SE (CFSE)
m-reMAIT cells (differentiation on days 18–23) were labeled with CFSE (1.0 µM) and cultured on OP9/DLL1 in αMEM containing 20% FBS (BioSera) and 1 ng/ml mouse IL-7. CFSE intensity in the cells was monitored daily by flow cytometry with 488 nm excitation and a bandpass filter of 530/30 nm. The proliferation rate of m-reMAIT cells was calculated from a logarithmical growth curve based on the intensity of CFSE.

## Detection of tyrosine-phosphorylated proteins
m-reMAIT cells (TCRβ+mMR1-tet+ cells, >95% based on the flow cytometric analysis) were suspended in 20% FBS containing αMEM ($1.0 \times 10^6$/ml) for the indicated time in the absence or presence of varying amounts of 5-OP-RU or unlabeled mMR1-tet. Cells were harvested by centrifugation at 600 × $g$ and lysed with lysis buffer (50 mM HEPES-NaOH pH 7.4, 150 mM NaCl, 1% [v/v] Triton-X100, 1 mM sodium vanadate, and 5 mM NaF) supplemented with ×1 protease and phosphatase inhibitor cocktail (FUJIFILM Wako) on ice for 60 min. Cell lysates were then centrifuged at 12,000 × $g$ at 4°C for 20 min, diluted with ×4 SDS sample buffer (240 mM Tris-HCl, pH 6.8, 8% SDS, 40% glycerol, 0.1% blue bromophenol, and 20% 2-mercaptoethanol), and heat-denatured at 97°C for 10 min. A cell lysate equivalent to $5 \times 10^5$ cells/lane was loaded onto a 10% SDS polyacrylamide gel (ePAGEL E-T10, Atto). After electrophoresis, proteins were blotted onto Immobilon-P (Millipore) with a semi-dry blotting apparatus (TRANS-BLOT SD, Bio-Rad). The membrane was blocked with TBS-T (50 mM Tris-HCl pH 7.5, 150 mM NaCl, and 0.05% [v/v] Tween-20) containing 5% BSA at 4°C for 18 hr. The membrane was incubated with the primary antibody in TBS-T containing 5% BSA (PY99: 1/1000; LAT: 1/1000; and β-actin: 1/2000 dilution) at room temperature for 60 min, then washed with TBS-T three times. The membrane was incubated with the secondary antibody (HRP-conjugated anti-mouse IgG, diluted 1/3000 in TBS-T containing 5% BSA) for 30 min and washed with TBS-T four times. Proteins and tyrosine-phosphorylated proteins were developed with SuperSignal Pico PLUS (Thermo Fisher) and visualized with a chemiluminometer (LuminoGraph 1, Atto).

## Flow cytometry

Cells were stained with the antibodies listed in the Key sources table. 7-AAD or Zombie Violet (BioLegend) was used to discriminate between live/dead cells. To stain the transcription factors PLZF and RORγt, cells stained with the surface markers were fixed and permeabilized with the Transcription Factor Buffer Set (BD Biosciences), and then stained with transcription factor antibodies. In CD107a staining, a fluorochrome- labeled CD107a antibody was added to the culture prior to the assay. Cells were analyzed with the MACSQuant cell analyzer (3 lasers, 10 parameters, Miltenyi Biotec) or the AttuneNxT acoustic focusing cytometer (4 lasers, 14 parameters, Thermo Fisher Scientific). Data were processed using FlowJo software (version 9 or 10, BD Biosciences). Cell sorting was performed using a FACSJazz cell sorter (two lasers, eight parameters, BD Biosciences).

## Cytokine and chemokine quantification

The quantification of cytokines and chemokines was performed with the LegendPlex mouse Th cytokine panel, mouse cytokine panel 2, and mouse proinflammatory chemokine panel according to the protocol provided by the manufacturer (BioLegend).

## Migration of m-reMAIT cells in syngeneic C57BL/6

Ly5.2-m-reMAIT cells (L7-1: $1.0 \times 10^7$ cells/mouse) were i.p. injected into C57BL/6 (Ly5.1) recipients, and the expression of markers in endogenous (Ly5.1) and exogenous MAIT cells (Ly5.2 m-reMAIT cells) was assessed using a combination of appropriate antibodies and mMR1-tet 7 and/or 14 days after their transfer unless otherwise indicated.

## RNA-sequencing analysis

Naïve m-reMAIT cells (L7-1, $1.0 \times 10^5$ cells), sort-purified liver m-reMAIT cells ($5.1 \times 10^3$ cells), spleen m-reMAIT cells ($1.4 \times 10^3$ cells), and LPL m-reMAIT cells ($2.3 \times 10^3$ cells) from the recipient mouse (Ly5.1) 14 days post-adoptive transfer were subjected to cDNA synthesis (SMART-seq v4, Takara) and RNA-sequencing (NovaSeq 6000, Illumina). Sort-purified endogenous MAIT cells (Ly5.1) from the corresponding organs ($1.2 \times 10^4$ cells for liver, $2.5 \times 10^4$ cells for spleen, and $1.4 \times 10^4$ cells for LPL) were also processed.

## Transcriptome analysis

FPKM was calculated from the RNA-seq read count data (121,806 genes) and logarithmically converted and normalized based on the expression level of the housekeeping genes with RUV function in R package (*Jacob et al., 2016*). Genes exhibiting a constant expression level across the samples were considered to be housekeeping (108 genes). PCA was calculated with 'prcomp' and visualized with 'ggplot2' in R package using the above data. Then the genes relevant to MAIT cell identity, function, and development were categorized, and the heatmap was created with 'pheatmap' in R package. Only differentially expressed genes (naïve m-reMAIT cell/endogenous MAIT cells or adoptively transferred m-reMAIT cells < 0.5 or > 2) are shown. Differentially expressed genes were analyzed with 'TCC' in R package (*Kadota et al., 2012*; *McCarthy et al., 2012*; *Sun et al., 2013*).

## Time course of MR1 expression in cancer cell lines upon the 5-OP-RU challenge

B16F10, EL4, LLC, RL♂1, and Yac-1 were incubated with 600 nM of 5-OP-RU. Thereafter, the geometric mean fluorescent intensity (gMFI) of MR1 on the cell surface was followed by flow cytometry. gMFI at the indicated time point relative to that at the nontreated state (time 0) was calculated and shown as relative gMFI.

## Tumor studies

LLC suspended in HBSS was intravenously (i.v.) inoculated ($3.0 \times 10^5$ cells/mouse) 6 days after the adoptive transfer of m-reMAIT cells ($1.0 \times 10^6$ cells/mouse unless otherwise indicated, i.p.) or HBSS alone. In the survival assay, mice were considered to be dead when they showed a humane end point, such as acute weight loss, hypothermia, and severe gait and/or consciousness disturbance, and the survival time of mice was plotted as a Kaplan–Meier curve. In the in situ tumor growth analysis, LLCs ($3.0 \times 10^5$ cells/mouse) were subcutaneously inoculated 6–8 days after the adoptive transfer of

m-reMAIT cells ($0.3 \times 10^6$, $1.0 \times 10^6$, and $3.0 \times 10^6$ cells/mouse, i.p.) or HBSS alone. Tumor sizes were measured with calipers every 3 days. Tumor volumes were calculated using the following formula: v = (tumor width)$^2$ × (tumor length)/2.

## Interaction between NK cells and m-reMAIT cells

NK cells were isolated from C57BL/6 mice (male and female) with the MojoSort mouse NK cell isolation kit (BioLegend) (purity >85% based on the flow cytometric analysis with NK1.1 and CD49b antibodies) and cocultured with m-reMAIT cells at 37°C in 5% $CO_2$ for 18 hr at the indicated ratio. In parallel, m-reMAIT cells and NK cells were cultured individually under the same conditions, and CD69 expression in NK cells (gated as NK1.1$^+$CD49b$^+$ cells) and m-reMAIT cells (gated as TCRβ$^+$mMR1-tet$^+$ cells) was analyzed using flow cytometry. Cytokines and chemokines released in the culture were measured with LegendPlex at the indicated ratio. In CD69 and CD107a expression analyses, the above column-isolated NK cells ($1 \times 10^5$ cells) and m-reMAIT cells ($1 \times 10^5$ cells, purity >95%) were cultured individually or cocultured in the absence or presence of the same number of Yac-1 at 37°C in 5% $CO_2$ for 18 hr. The percentages of CD69$^+$ cells and CD69$^+$CD107a$^+$ cells among NK cells and m-reMAIT cells were then measured using the MACSQuant flow cytometer.

## Semi-quantitative PCR

NK cells cocultured with m-reMAIT cells (NK cells/m-reMAIT cells ratio = 1) were sort-purified as NK1.1$^+$CD49b$^+$ NK cells and TCRβ$^+$mMR1-tet$^+$ m-reMAIT cells, respectively. RNA from each subpopulation was extracted with the RNeasy Mini kit (QIAGEN), and cDNA was synthesized with the first-strand cDNA synthesis kit (Thermo Fisher Scientific) and then subjected to semi-quantitative PCR. PCR was performed with CYBR Green reagent (Nippon Genetics) using the following program: at 95°C for 5 min (90°C for 15 s, 60°C for 60 s) × 50 cycles (Light/Cycler Nano, Roche). The primer sets used in this study are described in the Key resources table.

## Cytolytic activity

The cytolytic activities of NK cells and m-reMAIT cells were measured with Yac-1 and LLC as target cells. Yac-1 cells ($1.0 \times 10^4$ cells/assay) were labeled with CFSE 24 hr prior to the addition of effector cells, while LLC cells ($1.0 \times 10^4$ cells/assay) were not labeled. Target cells were then incubated with the effector cells at the indicated E/T ratio at 37°C in 5% $CO_2$ for 4 hr and 16 hr for Yac-1 cells and LLC cells, respectively. E/T ratios of 1/1, 1/3, and 1/10 in the assay using the combination of NK cells and m-reMAIT cells contained twice the number of effector cells than NK cells or m-reMAIT cells alone. Dying target cells were stained with the Zombie Flexible Viability kit according to the manufacturer's instructions. Percent lysis was calculated by the formula {(% of Zombie$^+$ cells among target cells in the presence of effector cells) − (% of Zombie$^+$ cells among target cells in the absence of effector cells)}/ 100 − (% of Zombie$^+$ cells among target cells in the absence of effector cells)/100.

## Quantification and statistical analysis

Statistical analyses were conducted using Prism 9 for macOS (GraphPad). The log-rank test was used for survival analyses between the two indicated groups. A two-way ANOVA was employed to assess the significance of differences among the various effector cells (NK cell, m-reMAIT cells, and NK cells plus m-reMAIT cells) in lysis assays.

### Data deposition

Sequencing data for *Trbv* of MAIT-iPSCs (*Table 1*) were deposited in DDJB under accession codes LC637403 for L3-1 clone, LC637404 for L7-1 clone, LC637405 for L11-1 clone, LC637406 for L15-1 clone, and LC637407 for L19-1 clone. RNA-seq data for naïve m-reMAIT cells (L7), spleen m-reMAIT cells, liver m-reMAIT cells, LPL m-reMAIT cells, endogenous MAIT cells in Ly5.1 spleen, endogenous MAIT cells in Ly5.1 liver, and endogenous MAIT cells in Ly5.1 LP were deposited under the accession number DRA013350 (hwakao-0001_Submission).

## Acknowledgements

We thank M Ohyama (Dokkyo Medical University) for technical help, Dr. Y Horihata (Department of Biochemistry, Dokkyo Medical University) for the structure analysis of synthesized 5-OP-RU, Y Machida

and T Tsukahara (Animal facility, Dokkyo Medical University) for tumor experiments, Drs. S Nishioka and M Ikawa (Biken, Osaka University, Osaka, Japan) for chimeric mice, Drs. M Nakanishi and M Ohtaka (Tokiwa Bio. Co., Ltd., Tsukuba, Japan) for KOSM302L, Dr. X Wang (Department of Pathology and Immunology, Washington University, MO, USA) for WT3, WT3/mMR1, CH27, and CH27/mMR1, Dr. T Seya (School of Medicine, Hokkaido University, Sapporo, Japan) for B16F10, the NIH Tetramer core facility (Emory University, GA, USA) for unlabeled and APC- and BV421-labeled mMR1-tetramers, and Dr. T Mistuyama (AIST, Tokyo) for preparing FPKM data and Advanced Animal Model Support for supporting chimeric mice (JSPS KAKENHI 16H06276).

## Additional information

### Funding

| Funder | Grant reference number | Author |
|---|---|---|
| Japan Society for the Promotion of Science | 17H03565 | Chie Sugimoto |
| Japan Society for the Promotion of Science | 26430084 | Chie Sugimoto |
| Japan Society for the Promotion of Science | 20K10435 | Hiroyoshi Fujita |
| Promotion and Mutual Aid Corporation for Private Schools of Japan | | Hiroshi Wakao |

The funders had no role in study design, data collection and interpretation, or the decision to submit the work for publication.

### Author contributions

Chie Sugimoto, Conceptualization, Data curation, Formal analysis, Funding acquisition, Investigation, Methodology, Project administration, Validation, Visualization, Writing - original draft; Yukie Murakami, Data curation, Investigation, Methodology; Eisuke Ishii, Investigation; Hiroyoshi Fujita, Data curation, Methodology, Project administration, Resources, Validation; Hiroshi Wakao, Conceptualization, Funding acquisition, Investigation, Project administration, Supervision, Validation, Writing - original draft, Writing - review and editing

### Author ORCIDs

Chie Sugimoto ⓘ http://orcid.org/0000-0002-0488-2931
Hiroshi Wakao ⓘ http://orcid.org/0000-0001-7851-0711

### Ethics

All mouse experiments were performed with approval from the Institutional Animal Care and Use Committee of Dokkyo Medical University (Permit Number: 1215).

### Decision letter and Author response

Decision letter https://doi.org/10.7554/eLife.70848.sa1
Author response https://doi.org/10.7554/eLife.70848.sa2

## Additional files

### Supplementary files

• Transparent reporting form

### Data availability

Sequencing data have been deposited in DDBJ under accession number LC637403 to LC637407. RNA-seq data for naïve m-reMAIT cells (L7), spleen m-reMAIT cells, liver m-reMAIT cells, LPL m-reMAIT cells , endogenous MAIT cells in Ly5.1 spleen, endogenous MAIT cells in Ly5.1

liver, and endogenous MAIT cells in Ly5.1 LP were deposited under the accession DRA013350 (hwakao-0001_Submission).

The following datasets were generated:

| Author(s) | Year | Dataset title | Dataset URL | Database and Identifier |
|---|---|---|---|---|
| Sugimoto C, Wakao H | 2021 | Reprogramming and redifferentiation of mucosal-associated invariant T cells reveal tumor inhibitory activity | http://getentry.ddbj. nig.ac.jp/getentry/na/ LC637403 | DNA Data Bank of Japan, LC637403 |
| Sugimoto C, Wakao H | 2021 | Reprogramming and redifferentiation of mucosal-associated invariant T cells reveal tumor inhibitory activity | http://getentry.ddbj. nig.ac.jp/getentry/na/ LC637404 | DNA Data Bank of Japan, LC637404 |
| Sugimoto C, Wakao H | 2021 | Reprogramming and redifferentiation of mucosal-associated invariant T cells reveal tumor inhibitory activity | http://getentry.ddbj. nig.ac.jp/getentry/na/ LC637405 | DNA Data Bank of Japan, LC637405 |
| Sugimoto C, Wakao H | 2021 | Reprogramming and redifferentiation of mucosal-associated invariant T cells reveal tumor inhibitory activity | http://getentry.ddbj. nig.ac.jp/getentry/na/ LC637406 | DNA Data Bank of Japan, LC637406 |
| Sugimoto C, Wakao H | 2021 | Reprogramming and redifferentiation of mucosal-associated invariant T cells reveal tumor inhibitory activity | http://getentry.ddbj. nig.ac.jp/getentry/na/ LC637407 | DNA Data Bank of Japan, LC637407 |
| Sugimoto C, Wakao H | 2022 | Reprogramming and redifferentiation of mucosal-associated invariant T cells reveal tumor inhibitory activity | https://ddbj.nig. ac.jp/resource/ sra-submission/ DRA013350 | DNA Data Bank of Japan, DRA013350 |

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

# Appendix 1

### Appendix 1—key resources table

| Reagent type (species) or resource | Designation | Source or reference | Identifiers | Additional information |
|---|---|---|---|---|
| Strain, strain background (*Mus musculus*) | C57BL/6NJcl | Clea Japan | | |
| Strain, strain background (*M. musculus*) | C57BL/6-Ly5.1 | Riken Bioresource Center | RRID:IMSR_RBRC00144 | |
| Strain, strain background (Sendai virus) | Sendai virus KOSM302L | Dr. Mahito Nakanishi (TOKIWA-Bio Inc) | | |
| Cell line (*M. musculus*) | L3-1, L7-1 to -8, L11-1 to -4, L15-1 to -3, L19-1 to -6 | This study | | Murine MAIT cell-derived iPSCs |
| Cell line (*M. musculus*) | Mouse embryonic fibroblast (MEF) | Oriental Yeast | Cat# KBL9284600 | |
| Cell line (*M. musculus*) | OP9 -DLL1 cell | Dr. Hiroshi Kawamoto (Kyoto University) | | |
| Cell line (*M. musculus*) | B16F10 cell | Dr. Tsukasa Seya (Hokkaido University) | RRID:CVCL_0159 | |
| Cell line (*M. musculus*) | Lewis lung carcinoma (LLC) | Riken Bioresource Center | Cat# RCB0558; RRID:CVCL_4358 | |
| Cell line (*M. musculus*) | EL4 | Riken Bioresource Center | Cat# RCB1641; RRID:CVCL_0255 | |
| Cell line (*M. musculus*) | RL-♂1 (Gloria) | Riken Bioresource Center | Cat# RCB2784; RRID:CVCL_C832 | |
| Cell line (*M. musculus*) | Yac-1 | Riken Bioresource Center | Cat# RCB1165; RRID:CVCL_2244 | |
| Cell line (*M. musculus*) | CH27 | Dr. X. Wang (Washington University, MO) | | |
| Cell line (*M. musculus*) | CH27/mMR1 | Dr. X. Wang (Washington University, MO) | | |
| Cell line (*M. musculus*) | WT3 | Dr. X. Wang (Washington University, MO) | | |
| Cell line (*M. musculus*) | WT3/mMR1 | Dr. X. Wang (Washington University, MO) | | |
| Antibody | Anti-mouse CD4 (RM4-4), PerCP/Cy5.5 (rat monoclonal) | BioLegend | Cat# 116012; RRID:AB_2563023 | Flow cytometry (1:100 in 50 µl reaction) |
| Antibody | Anti-mouse CD8α (53–6.7), APC/Cy7 (rat monoclonal) | BioLegend | Cat# 100714; RRID:AB_312753 | Flow cytometry (1:100 in 50 µl reaction) |
| Antibody | Anti-mouse CD25 (PC61), APC (rat monoclonal) | BioLegend | Cat# 102012; RRID:AB_312861 | Flow cytometry (1:100 in 50 µl reaction) |
| Antibody | Anti-mouse CD25 (PC61), BV421 (rat monoclonal) | BioLegend | Cat# 102043; RRID:AB_2562611 | Flow cytometry (1:100 in 50 µl reaction) |
| Antibody | Anti-mouse CD25 (PC61), BV711 (rat monoclonal) | BD Biosciences | Cat# 740652; RRID:AB_2740341 | Flow cytometry (1:100 in 50 µl reaction) |
| Antibody | Anti-mouse CD25 (PC61), PE (rat monoclonal) | BioLegend | Cat# 102007; RRID:AB_312856 | Flow cytometry (1:100 in 50 µl reaction) |
| Antibody | Anti-mouse/human CD44 (IM7), FITC (rat monoclonal) | BioLegend | Cat# 103006; RRID:AB_312957 | Flow cytometry (1:100 in 50 µl reaction) |
| Antibody | Anti-mouse CD45.1 (Ly5.1) (A20), BV605 (mouse monoclonal) | BioLegend | Cat# 110738; RRID:AB_2562565 | Flow cytometry (1:100 in 50 µl reaction) |
| Antibody | Anti-mouse CD45.2 (Ly5.2) (104), PE (mouse monoclonal) | BioLegend | Cat# 109807; RRID:AB_313444 | Flow cytometry (1:100 in 50 µl reaction) |
| Antibody | Anti-mouse/human CD45R/B220 (RA3-6B2), BV421 (rat monoclonal) | BioLegend | Cat# 103239; RRID:AB_10933424 | Flow cytometry (1:100 in 50 µl reaction) |

*Appendix 1 Continued on next page*

*Appendix 1 Continued*

| Reagent type (species) or resource | Designation | Source or reference | Identifiers | Additional information |
|---|---|---|---|---|
| Antibody | Anti-mouse/human CD45R/B220 (RA3-6B2), PE (rat monoclonal) | BioLegend | Cat# 103207; RRID:AB_312992 | Flow cytometry (1:100 in 50 µl reaction) |
| Antibody | Anti-mouse CD49b (DX5), PE (rat monoclonal) | BioLegend | Cat# 108907; RRID:AB_313414 | Flow cytometry (1:100 in 50 µl reaction) |
| Antibody | Anti-mouse CD69 (H1.2F3), APC/Cy7 (Armenian hamster monoclonal) | BioLegend | Cat# 104525; RRID:AB_10683447 | Flow cytometry (1:100 in 50 µl reaction) |
| Antibody | Anti-mouse CD69 (H1.2F3), PE (Armenian hamster monoclonal) | BD Biosciences | Cat# 553237; RRID:AB_394726 | Flow cytometry (1:100 in 50 µl reaction) |
| Antibody | Anti-mouse CD69 (H1.2F3), PE/Cy7 (Armenian hamster monoclonal) | BioLegend | Cat# 104511; RRID:AB_493565 | Flow cytometry (1:100 in 50 µl reaction) |
| Antibody | Anti-mouse CD107a (1D4B), BV421 (rat monoclonal) | BioLegend | Cat# 121618; RRID:AB_2749905 | Flow cytometry (1:100 in 50 µl reaction) |
| Antibody | Anti-mouse CD127 (IL-7Rα) (A7R34), PerCP/Cy5.5 (rat monoclonal) | BioLegend | Cat# 135021; RRID:AB_1937274 | Flow cytometry (1:100 in 50 µl reaction) |
| Antibody | Anti-mouse CD186 (CXCR6) (SAO51D1), PE (rat monoclonal) | BioLegend | Cat# 151103; RRID:AB_2566545 | Flow cytometry (1:100 in 50 µl reaction) |
| Antibody | Anti-mouse CD218a (IL-18Rα) (P3TUNYA), eFluor450 (rat monoclonal) | Thermo Fisher Scientific | Cat# 48-5183-80; RRID:AB_2574068 | Flow cytometry (1:100 in 50 µl reaction) |
| Antibody | Anti-mouse CD218a (IL-18Rα) (REA947), PE (mouse monoclonal) | Miltenyi Biotech | Cat# 130-115-704; RRID:AB_2727158 | Flow cytometry (1:100 in 50 µl reaction) |
| Antibody | Anti-mouse F4/80 (BM8), BV421 (rat monoclonal) | BioLegend | Cat# 123131; RRID:AB_10901171 | Flow cytometry (1:100 in 50 µl reaction) |
| Antibody | Anti-mouse F4/80 (BM8), PE (rat monoclonal) | BioLegend | Cat# 123109; RRID:AB_893498 | Flow cytometry (1:100 in 50 µl reaction) |
| Antibody | Anti-human/mouse/rat MR1 (26.5), APC (mouse monoclonal) | BioLegend | Cat# 361107; RRID:AB_2563193 | Flow cytometry (1:100 in 50 µl reaction) |
| Antibody | Anti-mouse NK1.1 (PK136), BV510 (mouse monoclonal) | BD Biosciences | Cat# 563096; RRID:AB_2738002 | Flow cytometry (1:100 in 50 µl reaction) |
| Antibody | Anti-mouse NK1.1 (PK136), FITC (mouse monoclonal) | BioLegend | Cat# 108706; RRID:AB_313393 | Flow cytometry (1:100 in 50 µl reaction) |
| Antibody | Anti-mouse PLZF (9E12), PE (Armenian hamster monoclonal) | BioLegend | Cat# 145803; RRID:AB_2561966 | Flow cytometry (1:100 in 50 µl reaction) |
| Antibody | Anti-mouse RORγt (Q31-378), BV421 (mouse monoclonal) | BD Biosciences | Cat# 562894; RRID:AB_2687545 | Flow cytometry (1:100 in 50 µl reaction) |
| Antibody | Anti-mouse TCR Vβ6 (RR4-7), PE (mouse monoclonal) | BioLegend | Cat# 140003; RRID:AB_10640727 | Flow cytometry (1:100 in 50 µl reaction) |
| Antibody | Anti-mouse TCR Vβ8.1, 8.2 (MR5-2), PE (mouse monoclonal) | BioLegend | Cat# 140103; RRID:AB_10641144 | Flow cytometry (1:100 in 50 µl reaction) |
| Antibody | Anti-mouse TCRβ (H57-597), BV605 (Armenian hamster monoclonal) | BioLegend | Cat# 109241; RRID:AB_2629563 | Flow cytometry (1:100 in 50 µl reaction) |
| Antibody | Anti-mouse TCRβ (H57-597), PE/Cy7 (Armenian hamster monoclonal) | BioLegend | Cat# 109222; RRID:AB_893625 | Flow cytometry (1:100 in 50 µl reaction) |
| Antibody | Anti-human/mouse/rat MR1 (26.5), purified (mouse monoclonal) | BioLegend | Cat# 361110; RRID:AB_2801000 | Blocking assay (10 µg/ml) |
| Antibody | Mouse IgG2a, k isotype control (MOPC-173), purified (mouse monoclonal) | BioLegend | Cat# 400264; RRID:AB_11148947 | Blocking assay (10 µg/ml) |
| Antibody | Anti-human/mouse/rat β-actin (2F1-1), purified (mouse monoclonal) | BioLegend | Cat# 643802; RRID:AB_2223199 | Western blotting (1:2000 in 5 ml reaction) |

*Appendix 1 Continued*

| Reagent type (species) or resource | Designation | Source or reference | Identifiers | Additional information |
|---|---|---|---|---|
| Antibody | Anti-human/mouse LAT (11B.12), purified (mouse monoclonal) | Santa Cruz | Cat# sc-53550; RRID:AB_784283 | Western blotting (1:1000 in 5 ml reaction) |
| Antibody | Anti-phosphotyrosine (PY99), purified (mouse monoclonal) | Santa Cruz | Cat# sc-7020; RRID:AB_628123 | Western blotting (1:1000 in 5 ml reaction) |
| Antibody | Anti-mouse/rat Asialo GM1 (rabbit polyclonal) | FUJIFILM Wako | Cat# 014-09801 | In vivo NK depletion (50 μg/mouse) |
| Sequence-based reagent | ADV19 | This study | PCR primer (detection of rearranged TCRVα19-Jα33) | 5'-TCAACTGCACAT ACAGCACCTC-3' |
| Sequence-based reagent | AJ33 | This study | PCR primer (detection of rearranged TCRVα19-Jα33) | 5'-CATGCATTATTCA GCCAGTGCCTTCT-3' |
| Sequence-based reagent | TRAV19-F | This study | PCR primer (Southern blot probe synthesis) | 5'-CCTGGACCACATG GAAGCATGGC-3' |
| Sequence-based reagent | TRAV19-R | This study | PCR primer (Southern blot probe synthesis) | 5'-CCCAGAGCC CCAGATCAAC-3' |
| Sequence-based reagent | TRBV13 | This study | PCR primer (identification of TCRβ repertoire) | 5'-GTACTGGTAT CGGCAGGAC-3' |
| Sequence-based reagent | TRBV19 | This study | PCR primer (identification of TCRβ repertoire) | 5'-GGTACCGAC AGGATTCAG-3' |
| Sequence-based reagent | TRBC-Rev | This study | PCR primer (identification of TCRβ repertoire) | 5'-GGGTAGCCT TTTGTTTGTTTG-3' |
| Sequence-based reagent | *Gapdh*-F | This study | PCR primer (semi-quantitative PCR) | 5'-CATCACTGCCAC CCAGAAGACTG-3' |
| Sequence-based reagent | *Gapdh*-R | This study | PCR primer (semi-quantitative PCR) | 5'-ATGCCAGTGAGC TTCCCGTTCAG-3' |
| Sequence-based reagent | *Tnf*-F | This study | PCR primer (semi-quantitative PCR) | 5'-CCACCACGC TCTTCTGTCTAC-3' |
| Sequence-based reagent | *Tnf*-R | This study | PCR primer (semi-quantitative PCR) | 5'-AGGGTCTGG GCCATAGAACT-3' |
| Sequence-based reagent | *Ifng*-F | This study | PCR primer (semi-quantitative PCR) | 5'-AAAGAGATAAT CTGGCTCTGC-3' |
| Sequence-based reagent | *Ifng*-R | This study | PCR primer (semi-quantitative PCR) | 5'-GCTCTGAGAC AATGAACGCT-3' |
| Sequence-based reagent | *Grza*-F | This study | PCR primer (semi-quantitative PCR) | 5'-GGTGGAAAG GACTCCTGCAA-3' |
| Sequence-based reagent | *Grza*-R | This study | PCR primer (semi-quantitative PCR) | 5'-GCCTCGCAA AATACCATCACA-3' |
| Sequence-based reagent | *Grzb*-F | This study | PCR primer (semi-quantitative PCR) | 5'-ACTCTTGACG CTGGGACCTA-3' |
| Sequence-based reagent | *Grzb*-R | This study | PCR primer (semi-quantitative PCR) | 5'-AGTGGGGCT TGACTTCATGT-3' |
| Sequence-based reagent | *Grzk*-F | This study | PCR primer (semi-quantitative PCR) | 5'-AAGCTTCGCACT GCTGCAGAACT-3' |
| Sequence-based reagent | *Grzk*-R | This study | PCR primer (semi-quantitative PCR) | 5'-TAACAGATCTGG CTTGGTGGTTCC-3' |
| Sequence-based reagent | *Prf1*-F | This study | PCR primer (semi-quantitative PCR) | 5'-CTCTCGAAGTG TTGGATACAG-3' |
| Sequence-based reagent | *Prf1*-R | This study | PCR primer (semi-quantitative PCR) | 5'-GACACAAACGTG ATTCAAATCC-3' |
| Sequence-based reagent | *Fasl*-F | This study | PCR primer (semi-quantitative PCR) | 5'-GAAGGAACTGGC AGAACTCCGT-3' |
| Sequence-based reagent | *Fasl*-R | This study | PCR primer (semi-quantitative PCR) | 5'-GCCACACTCCT CGGCTCTTTTT-3' |
| Sequence-based reagent | *Tnfsf10*-F | This study | PCR primer (semi-quantitative PCR) | 5'-GGAAGACCTCAG AAAGTGGCAG-3' |
| Sequence-based reagent | *Tnfsf10*-R | This study | PCR primer (semi-quantitative PCR) | 5'-TTTCCGAGAG GACTCCCAGGAT-3' |

*Appendix 1 Continued*

| Reagent type (species) or resource | Designation | Source or reference | Identifiers | Additional information |
|---|---|---|---|---|
| Sequence-based reagent | *Il6*-F | This study | PCR primer (semi-quantitative PCR) | 5′-TACCACTTCACA AGTCGGAGGC-3′ |
| Sequence-based reagent | *Il6*-R | This study | PCR primer (semi-quantitative PCR) | 5′-CTGCAAGTGCAT CATCGTTGTTC-3′ |
| Sequence-based reagent | *Il17a*-F | This study | PCR primer (semi-quantitative PCR) | 5′-CAGACTACCTC AACCGTTCCAC-3′ |
| Sequence-based reagent | *Il17a*-R | This study | PCR primer (semi-quantitative PCR) | 5′-TCCAGCTTTCC CTCCGCATTGA-3′ |
| Sequence-based reagent | *Il22*-F | This study | PCR primer (semi-quantitative PCR) | 5′-GCTTGAGGTGT CCAACTTCCAG-3′ |
| Sequence-based reagent | *Il22*-R | This study | PCR primer (semi-quantitative PCR) | 5′-ACTCCTCGGAA CAGTTTCTCCC-3′ |
| Sequence-based reagent | *Ccl5*-F | This study | PCR primer (semi-quantitative PCR) | 5′-CCTGCTGCTTT GCCTACCTCTC-3′ |
| Sequence-based reagent | *Ccl5*-R | This study | PCR primer (semi-quantitative PCR) | 5′-ACACACTTGGC GGTTCCTTCGA-3′ |
| Sequence-based reagent | *Ccl3*-F | This study | PCR primer (semi-quantitative PCR) | 5′-ACTGCCTGCTGC TTCTCCTACA-3′ |
| Sequence-based reagent | *Ccl3*-R | This study | PCR primer (semi-quantitative PCR) | 5′-ATGACACCTGGC TGGGAGCAAA-3′ |
| Sequence-based reagent | *Ccl4*-F | This study | PCR primer (semi-quantitative PCR) | 5′-ACCCTCCCACT TCCTGCTGTTT-3′ |
| Sequence-based reagent | *Ccl4*-R | This study | PCR primer (semi-quantitative PCR) | 5′-CTGTCTGCCTC TTTTGGTCAGG-3′ |
| Sequence-based reagent | *Ccl22*-F | This study | PCR primer (semi-quantitative PCR) | 5′-GTGGAAGACAG TATCTGCTGCC-3′ |
| Sequence-based reagent | *Ccl22*-R | This study | PCR primer (semi-quantitative PCR) | 5′-AGGCTTGCGGC AGGATTTTGAG-3′ |
| Peptide, recombinant protein | Mouse MR1 5-OP-RU tetramer, APC-labeled | NIH Tetramer Core Facility | | Flow cytometry (1:1,000 in 50 µl reaction) |
| Peptide, recombinant protein | Mouse MR1 5-OP-RU tetramer, BV421-labeled | NIH Tetramer Core Facility | | Flow cytometry (1:1,000 in 50 µl reaction) |
| Peptide, recombinant protein | Mouse MR1 5-OP-RU tetramer, unlabeled | NIH Tetramer Core Facility | | MAIT cell stimulation (1:100 to 1:100,000 in 100 µl reaction) |
| Peptide, recombinant protein | Mouse MR1 6-FP tetramer, APC-labeled | NIH Tetramer Core Facility | | Flow cytometry (1:1,000 in 50 µl reaction) |
| Peptide, recombinant protein | Mouse MR1 6-FP tetramer, BV421-labeled | NIH Tetramer Core Facility | | Flow cytometry (1:1,000 in 50 µl reaction) |
| Peptide, recombinant protein | Mouse MR1 6-FP tetramer, unlabeled | NIH Tetramer Core Facility | | MAIT cell stimulation (1:100 to 1:100,000 in 100 µl reaction) |
| Commercial assay or kit | LEGENDPlex mouse Th cytokine panel | BioLegend | Cat# 740741 | |
| Commercial assay or kit | LEGENDPlex mouse cytokine panel 2 | BioLegend | Cat# 740134 | |
| Commercial assay or kit | LEGENDPlex mouse proinflammatory chemokine panel | BioLegend | Cat# 740451 | |
| Commercial assay or kit | MojoSort mouse NK cell isolation kit | BioLegend | Cat# 480050 | |
| Commercial assay or kit | Vybrant CFDA SE cell tracer kit | Thermo Fisher Scientific | Cat# V12883 | |
| Commercial assay or kit | Zombie Violet Flexible Viability kit | BioLegend | Cat# 423113 | |
| Chemical compound, drug | 5-Amino-4-D-ribitylaminouracil Dihydrochloride | Toronto Research Chemicals | Cat# A629245 | |
| Software, algorithm | FlowJo software v9 and v10 | BD Biosciences | RRID:SCR_008520 | |
| Software, algorithm | Prism 9 for macOS | GraphPad Software | RRID:SCR_002798 | |

