## [Editor Report]

This study exploits a reprogramming approach to study MAIT cells that can partially overcome the current technological limitations. Overall, the reviewers found that this work is provocative and novel.

---

## [Decision Letter]

**Decision letter after peer review:**

Thank you for submitting your article "Reprogramming and redifferentiation of mucosal–associated invariant T cells reveals tumor inhibitory activity" for consideration by *eLife*. Your article has been reviewed by 3 peer reviewers, one of whom is a member of our Board of Reviewing Editors, and the evaluation has been overseen by Carla Rothlin as the Senior Editor. The reviewers have opted to remain anonymous.

Essential revisions:

1) Since transcriptomic and epigenetic events is affected by reprogramming and play important role in cell identity. It would be important to determine the transcriptome and chromatin accessibility in the reprogrammed MAITs and real MAITs from B6 mice. The comparisons will further strengthen the conclusion and allows the author more precisely interpret their results.

2) Is the lack of anti–tumor activity against subcutaneously injected LLC truly due to lack of reMAIT–induced anti–tumor immunity or potentially due to the absence or inefficient migration and residency of reMAIT cells into the skin? In this regard, a detailed analysis to elucidate the differences is needed.

*Reviewer #1 (Recommendations for the authors):*

Understanding the biology of MAITs is a challenging for immunologists even though this cell type represent interesting functions in a variety of diseases. In this manuscript, the authors present an exciting approach with compelling data to indicate the power of reprogramming on studying MAITs as well as the therapeutic potential on harnessing MAITs for anti-tumor responses. Overall, it is well performed and aims to address interesting and important questions with excellent approaches. Although I find this manuscript interesting, I do have some concerns on the reprogramming approaches the authors applied on the detailed cellular identity. Since transcriptomic and epigenetic events are affected by reprogramming and play important role in cell identity, it would be important to determine the transcriptome and chromatin accessibility in the reprogrammed MAITs and real MAITs from B6 mice. The comparisons will further strengthen the conclusion and allows the author more precisely interpret their results.

*Reviewer #2 (Recommendations for the authors):*

a) Figure 3 contains a panel (E) that is the same as Figure 4 (H), which is the actual one used by the authors and discussed in the manuscript. The authors should be more careful to avoid this.

b) The authors demonstrate in vitro that exposure to 5–OP–RU, an agonist for MAIT cells, does not significantly upregulate MR1 on LLC cell line compared to B16F10 in Figure 3A. This was consistent with their observation that reMAIT cells do not exhibit cytolytic activity against LLC in vitro (Figure 4). However, reMAIT cells showed comparable cytolytic activity against NK cell–sensitive Yac–1 cell line, which showed a similar expression pattern of MR1 as LLC upon exposure to 5–OP–RU.

i. Do Yac–1 cells express molecules that can commonly activate NK cells and reMAIT cells?

ii. Is this expected of any MAIT cells, such as TCR transgenic MAIT cells that are commercially available?

iii. What is their MR1 expression pattern once injected into mice?

c) The authors demonstrated that adoptive transfer of reMAIT cells could significantly improve the survival of mice challenged with i.v. injection of LLC, an experimental model of lung metastasis. However, reMAIT cells did not enhance anti–tumor immunity against the same LLC when it was injected subcutaneously. From such observations, the authors concluded that reMAIT cells are inhibiting the metastasis of LLC rather than inducing anti–tumor immunity against them.

i. Is the lack of anti–tumor activity against subcutaneously injected LLC truly due to lack of reMAIT–induced anti–tumor immunity or potentially due to the absence or inefficient migration and residency of reMAIT cells into the skin? While the authors interrogated the distribution of transferred reMAIT cells in various organs, they did not show the skin where the frequency of MAIT cells among αβT cells is the highest as previously reported by the Belkaid group (PMID: 31649166). It is crucial to show the abundance of the transferred reMAIT cells in the organs relevant to the disease models used in the study.

ii. Perhaps using a luciferase–expressing LLC cell line allows monitoring the metastasis and growth of i.v. injected LLC in the presence of reMAIT cells as well.

iii. Considering the clinical potential of these iPSC–derived reMAIT cells, do reMAIT cells exhibit similar protective functions when transferred at a therapeutic time point rather than prophylactic transfer before the tumor challenge?

d) Given that the transferred reMAIT cells showed migrations into various tissues as demonstrated in Figure 2, can reMAIT cells be generated from MAIT cells isolated from other organs than the lung, such as skin or intestines, and still exhibit a similar protective role against the lung tumor metastasis model?

e) The authors nicely demonstrated that the tumor inhibitory function of reMAIT cells requires NK cells.

i. However, the transition from the observations in Figure 3 to NK–reMAIT interaction seems a little haste. It may be helpful if the authors could discuss in more depth the relevance between NK cells and MAIT cells in any context, whether cancer immunology or others, before presenting the data in Figure 4.

ii. Is the NK–induced activation of reMAIT cells cell–cell contact–dependent? If so, is it possible to show the proximity of NK cells and reMAIT cells in the TME by using microscopy to investigate the existence of their communications in vivo?

*Reviewer #3 (Recommendations for the authors):*

Overall this work is well performed and the conclusions are supported by the data. The manuscript is clear and easy to read, which is appreciated. However, considering the in vitro differentiated system and despite being an original model, I am a bit concerned how this could be translated into physiological settings regarding the real role of MAIT cells in cancer. In order to strengthen these claims, I think the study could really benefit from additional comparison between endogenous vs. regenerated MAIT cells (are all the surface markers and transcription factors conserved?) and/or in vitro activation assays including microbial–associated antigens to mimic the tumor microenvironment.

Specific comments:

– Can the authors elaborate more on the differences found between 5–OP–RU vs. mMR1–tet induced stimulations? It is briefly mentioned in the discussion but more mechanistic hypothesis could be nice.

– I think the discrepancies found in different human studies could be more discussed, because this remains one fundamental question in our poor understanding of how MAIT cells behave in cancer. For instance, they could comment on the differences found across cancer types, since one model only has been employed here for in vivo experiments. Would the authors expect to get the same response in a different tumor?

– Figure 1D–E: Activation of MAIT cells upon 5–OP–RU or mMR1–tet challenges are measured through CD69. I wonder if murine MAIT cells will express CD39 upon TCR stimulation as shown in humans (Li et al., Cell Rep Med, 2020)?

– The Figure 2 is quite interesting: can the authors discuss the distribution of regenerated MAIT cells in the different tissues upon adoptive transfer? Since they assessed CD44 expression at two time points, what is the frequency of reMAIT cells at these two timings? It would have been interesting to track/see how they populate the different organs.

– Figure 3D: can the authors show the tumor volume in the IV LLC model? I think it is important discuss the differences found between the IV and subcutaneous model. If reMAIT cells could not inhibit tumor growth in situ, the authors should at least change the conclusions in the abstract that is misleading, or test additional tumor models.

– It is a bit regretful that the authors generated this tool to explore interactions between MAIT vs. immune cells and only NK cells were studied. Since the role of NKT cells are quite well described in cancer and share a lot in terms of developmental and transcriptional pathways with MAIT cells, I wonder how they are affected in this system?

[Editors' note: further revisions were suggested prior to acceptance, as described below.]

Thank you for submitting your article "Reprogramming and redifferentiation of mucosal–associated invariant T cells reveals tumor inhibitory activity" for consideration by *eLife*. Your article has been reviewed by 2 peer reviewers, one of whom is a member of our Board of Reviewing Editors, and the evaluation has been overseen by Carla Rothlin as the Senior Editor. The reviewers have opted to remain anonymous.

Summary:

This study utilise a state–of–art approach, re–differentiation of cells, to extend our understanding in MAIT cells. By using this approach, we will be able to gain more insight on MAIT cells with less concerns on cell number issue. Overall, reviewers all feel this study provide an interesting advance for MAIT cells and presents a strong platform for future works.

Essential revisions:

1) Please include in the discussion the limitations raised by in vitro redifferentiation of the MAIT cells in interpreting the data.

2) Several redundancies and typos are still found in the manuscript, a careful reading will be needed before publication.

*Reviewer #2 (Recommendations for the authors):*

The authors replied to comments in a satisfactory manner and provided new data that improved the manuscript. The manuscript is now suitable for publication.

*Reviewer #3 (Recommendations for the authors):*

Overall, the authors have provided additional analysis to help answer some of the key questions and have appropriately addressed most of the comments.

– Please include in the discussion the limitations raised by in vitro redifferentiation of the MAIT cells in interpreting the data, as the transcriptomic analysis shows and confirms a clear different pattern for endogenous and redifferentiated cells.

– I would suggest the authors to specify in the abstract that reMAIT cells inhibit tumour growth in a lung metastasis model.

– Several redundancies and typos are still found in the manuscript, a careful reading will be needed before publication.

---

## [Author Response]

Essential revisions:1) Since transcriptomic and epigenetic events is affected by reprogramming and play important role in cell identity. It would be important to determine the transcriptome and chromatin accessibility in the reprogrammed MAITs and real MAITs from B6 mice. The comparisons will further strengthen the conclusion and allows the author more precisely interpret their results.2) Is the lack of anti–tumor activity against subcutaneously injected LLC truly due to lack of reMAIT–induced anti–tumor immunity or potentially due to the absence or inefficient migration and residency of reMAIT cells into the skin? In this regard, a detailed analysis to elucidate the differences is needed.

As you will see in the response to the reviewer concerns, we have performed transcriptome analyses and addressed the migration of m-reMAIT cells in the skin. We hope that these additional data and revision of the manuscript based on the reviewer concerns and comments will make it possible to accept our paper for publication in the journal.

Reviewer #1 (Recommendations for the authors):Understanding the biology of MAITs is a challenging for immunologists even though this cell type represent interesting functions in a variety of diseases. In this manuscript, the authors present an exciting approach with compelling data to indicate the power of reprogramming on studying MAITs as well as the therapeutic potential on harnessing MAITs for anti-tumor responses. Overall, it is well performed and aims to address interesting and important questions with excellent approaches. Although I find this manuscript interesting, I do have some concerns on the reprogramming approaches the authors applied on the detailed cellular identity. Since transcriptomic and epigenetic events are affected by reprogramming and play important role in cell identity, it would be important to determine the transcriptome and chromatin accessibility in the reprogrammed MAITs and real MAITs from B6 mice. The comparisons will further strengthen the conclusion and allows the author more precisely interpret their results.

Thank you for your appreciation on our manuscript. While we agree that determination of the transcriptome and chromatin accessibility in the reprogrammed MAITs and real MAITs from B6 mice is of importance, these analyses are technically challenging due to the rarity of the endogenous MAIT cells as exemplified in the previous study (Hinks et al., Cell Reports, 28 3249-3262, 2019). Nonetheless, we have now addressed the transcriptome in reprogrammed naïve MAITs, in reprogrammed MAIT cells after adoptive transfer, and in the endogenous MAIT cells in C57BL/6 mice. It turned out that the transcriptome of naïve m-reMAIT cells clustered upon adoptive transfer, and has become reminiscent to that of endogenous MAIT cells from the lamina propria lymphocytes concomitant with upregulation of transcripts relevant to MAIT cell identity and function. Unfortunately, we could not address the chromatin accessibility. We hope that these additional data would further strengthen the quality of our manuscript and satisfy the reviewers critiques.

Reviewer #2 (Recommendations for the authors):a) Figure 3 contains a panel (E) that is the same as Figure 4 (H), which is the actual one used by the authors and discussed in the manuscript. The authors should be more careful to avoid this.

We have deleted the duplicated figure. We sincerely apologize the inconvenience.

b) The authors demonstrate in vitro that exposure to 5–OP–RU, an agonist for MAIT cells, does not significantly upregulate MR1 on LLC cell line compared to B16F10 in Figure 3A. This was consistent with their observation that reMAIT cells do not exhibit cytolytic activity against LLC in vitro (Figure 4). However, reMAIT cells showed comparable cytolytic activity against NK cell–sensitive Yac–1 cell line, which showed a similar expression pattern of MR1 as LLC upon exposure to 5–OP–RU.i. Do Yac–1 cells express molecules that can commonly activate NK cells and reMAIT cells?

Unfortunately, we did not analyze the cell surface molecules in Yac-1. While it is known that Yac-1 expresses NKG2D ligands, important for recognition by NKG2D in anti-tumor activity (Nausch and Cerwenka, Oncogene 27 5944-5958 2008), our flow cytometry analysis failed to detect NKG2D in naïve reMAIT cells. However, transcriptome analysis has revealed that the transcript for NKG2D was upregulated in reMAIT cells upon adoptive transfer (data will be available upon request). This strongly indicated that reMAIT cells may be activated by NKG2D ligands present in Yac-1upon contact with NK cells, which in turn led to enhanced cytolytic activity in vivo. Thus, we could speculate that ligands for NKG2D in Yac-1 activated both NK cells and reMAIT cells in vitro.

ii. Is this expected of any MAIT cells, such as TCR transgenic MAIT cells that are commercially available?

Unfortunately, we do not have TCR transgenic MAIT cells in our facility. Thus, we could not answer.

iii. What is their MR1 expression pattern once injected into mice?

We could not measure it. Since LLC was injected intravenously, it is technically highly challenging to address such a question.

c) The authors demonstrated that adoptive transfer of reMAIT cells could significantly improve the survival of mice challenged with i.v. injection of LLC, an experimental model of lung metastasis. However, reMAIT cells did not enhance anti–tumor immunity against the same LLC when it was injected subcutaneously. From such observations, the authors concluded that reMAIT cells are inhibiting the metastasis of LLC rather than inducing anti–tumor immunity against them.i. Is the lack of anti–tumor activity against subcutaneously injected LLC truly due to lack of reMAIT–induced anti–tumor immunity or potentially due to the absence or inefficient migration and residency of reMAIT cells into the skin? While the authors interrogated the distribution of transferred reMAIT cells in various organs, they did not show the skin where the frequency of MAIT cells among αβT cells is the highest as previously reported by the Belkaid group (PMID: 31649166). It is crucial to show the abundance of the transferred reMAIT cells in the organs relevant to the disease models used in the study.

Thank you for your thoughtful comments. While it is true that the skin contains MAIT cells, the absolute number of MAIT cells is an order of ~10^2^ cells/100 mm^3^. Concomitant with the published observation, we observed few reMAIT cell in the skin close to the tumor (please see Figure3—figure supplement 1). From these results, it is reasonable to conclude that the lack of anti-tumor activity against subcutaneously injected LLC is due to the inefficient migration and residency of reMAIT cells in the skin, rather than the lack of anti-tumor activity of reMAIT cells per se. It is also noteworthy that such poor anti-tumor activity by m-reMAIT cells is, in part, due to poor infiltration into the tumor. Indeed, LLC is known to harbor little tumor-infiltrating lymphocytes (Li, H.Y. et al., Cancer Immunol.Res. 5, 767–777 2017).

ii. Perhaps using a luciferase–expressing LLC cell line allows monitoring the metastasis and growth of i.v. injected LLC in the presence of reMAIT cells as well.

Thank you for your comments. In this study, we focused our study on the role of reMAIT cells in mouse survival rather than in metastasis of LLC per se. Future study could address this point.

iii. Considering the clinical potential of these iPSC–derived reMAIT cells, do reMAIT cells exhibit similar protective functions when transferred at a therapeutic time point rather than prophylactic transfer before the tumor challenge?

Thank you for your thoughtful comments. We agree that this is a subject to be addressed in the near future for translational medicine.

d) Given that the transferred reMAIT cells showed migrations into various tissues as demonstrated in Figure 2, can reMAIT cells be generated from MAIT cells isolated from other organs than the lung, such as skin or intestines, and still exhibit a similar protective role against the lung tumor metastasis model?

We could not address it. Our study has revealed that the source of MAIT cells was of prime importance for successful reprogramming, as we could not succeed in obtaining iPSCs from MAIT cells prepared from other organs under the same conditions.

e) The authors nicely demonstrated that the tumor inhibitory function of reMAIT cells requires NK cells.i. However, the transition from the observations in Figure 3 to NK–reMAIT interaction seems a little haste. It may be helpful if the authors could discuss in more depth the relevance between NK cells and MAIT cells in any context, whether cancer immunology or others, before presenting the data in Figure 4.

Thank you for your suggestion. We have added “NK cells are an essential innate sentinel in tumor immunosurveillance, and previous studies have shown contradicting results with regard to the role of MAIT cells on NK cells in tumor immunity. Yan et al. demonstrated that MAIT cells promote tumor growth by suppressing the activity of NK cells and T cells, while others have suggested that activating MAIT cells in vivo strengthens anti-tumor activity concomitant with enhanced NK cell response (Yan et al., 2020a)(Petley et al., 2021). We thus addressed whether and how m-reMAIT cells and NK cells mutually affected their functions in tumor immunity. “in the text (p13-14).

ii. Is the NK–induced activation of reMAIT cells cell–cell contact–dependent? If so, is it possible to show the proximity of NK cells and reMAIT cells in the TME by using microscopy to investigate the existence of their communications in vivo?

Thank you for your thoughtful suggestion. From our preliminary experiments, we would infer that the direct contact between reMAIT cells and NK cells might be essential for reMAIT cells to be activated. Regarding the microscopic observation, we are a little bit afraid that our LLC system is not suitable for this purpose, as LLC is known to harbor little tumor infiltrating lymphocytes (Li, H.Y. et al., Cancer Immunol.Res. 5, 767–777 2017). It would be better if we could find a TME that contains both NK cells and reMAIT cells to observe such an interaction.

Reviewer #3 (Recommendations for the authors):Overall, this work is well performed and the conclusions are supported by the data. The manuscript is clear and easy to read, which is appreciated. However, considering the in vitro differentiated system and despite being an original model, I am a bit concerned how this could be translated into physiological settings regarding the real role of MAIT cells in cancer. In order to strengthen these claims, I think the study could really benefit from additional comparison between endogenous vs. regenerated MAIT cells (are all the surface markers and transcription factors conserved?) and/or in vitro activation assays including microbial–associated antigens to mimic the tumor microenvironment.Specific comments:– Can the authors elaborate more on the differences found between 5–OP–RU vs. mMR1–tet induced stimulations? It is briefly mentioned in the discussion but more mechanistic hypothesis could be nice.

Thank you for your suggestion. According to the suggestion, we have added the following sentences in the discussion “In contrast, mMR1-tet might have activated signaling pathways pertinent to Th-17 activation as evidenced with the production of Th-17 cytokines such as IL-17F, IL-22, and IL-23 concomitant with inflammatory chemokines such as RANTES, CCL2, CXCL1, and CXCL6 together with TNF-α (Figure 1F). These data suggested that mMR1-tet signals, in part, through NF-κB, while 5-OP-RU fails to do so“ (p18-19).

– I think the discrepancies found in different human studies could be more discussed, because this remains one fundamental question in our poor understanding of how MAIT cells behave in cancer. For instance, they could comment on the differences found across cancer types, since one model only has been employed here for in vivo experiments. Would the authors expect to get the same response in a different tumor?

Thank you for your thoughtful suggestion. We have now added more information on MAIT cells across different human cancers in Introduction (p3-4).

“With respect to the response in a different tumor, we did not have enough data to discuss more. Please note, however, that when we have applied the same technique (adoptive transfer of reMAIT cells into syngeneic mice before tumor inoculation) to B16F10 melanoma metastasis model, we have obtained a similar positive effect on the overall survival.

– Figure 1D–E: Activation of MAIT cells upon 5–OP–RU or mMR1–tet challenges are measured through CD69. I wonder if murine MAIT cells will express CD39 upon TCR stimulation as shown in humans (Li et al., Cell Rep Med, 2020)?

We could not detect the expression of CD39 in endogenous MAIT cells in C57BL/6.

– The Figure 2 is quite interesting: can the authors discuss the distribution of regenerated MAIT cells in the different tissues upon adoptive transfer? Since they assessed CD44 expression at two time points, what is the frequency of reMAIT cells at these two timings? It would have been interesting to track/see how they populate the different organs.

Thank you for your comments on the distribution of regenerated MAIT cells in the recipient mice. We have provided the data relevant to this concern (please see Figure 2D, supplemented in the revised version). It seemed that the frequency of reMAIT cells in the primary lymphoid tissues (thymus and bone marrow) decreases, whereas that in other tissues including the secondary lymphoid tissues tends to increase over time.

– Figure 3D: can the authors show the tumor volume in the IV LLC model? I think it is important discuss the differences found between the IV and subcutaneous model. If reMAIT cells could not inhibit tumor growth in situ, the authors should at least change the conclusions in the abstract that is misleading, or test additional tumor models.

Thank you for your fruitful comments. When LLC was injected intravenously, we could not measure the tumor volume, as LLC enters circulation. In contrast, we could measure the tumor volume of LLC when LLC was injected subcutaneously. As shown in the revised manuscript, we could show that there existed only few reMAIT cell in the skin covering the tumor, implying inefficient migration of reMAIT cells in the skin or into the tumor.

– It is a bit regretful that the authors generated this tool to explore interactions between MAIT vs. immune cells and only NK cells were studied. Since the role of NKT cells are quite well described in cancer and share a lot in terms of developmental and transcriptional pathways with MAIT cells, I wonder how they are affected in this system?

Thank you for your thoughtful suggestion. We agree with the reviewer in that iNKT cells and MAIT cells share a lot in terms of developmental and transcriptional pathways. To answer this, we now have compared the transcriptomes among naïve reMAIT cells (regenerated MAIT cells from iPSCs in the culture dish), reMAIT cells recovered after adoptive transfer into C57Bl/6 mice, and endogenous MAIT cells from congeneic C57Bl/6 mice. The results indicated that expression of the transcripts for transcription factors and effector molecules relevant to MAIT cell identity and function was upregulated upon adoptive transfer and the expression level has become close to that of endogenous ones (please see Figure 2—figure supplement 1-5). However, we did not address the transcriptomes in iNKT cells.

[Editors' note: further revisions were suggested prior to acceptance, as described below.]

Essential revisions:1) Please include in the discussion the limitations raised by in vitro redifferentiation of the MAIT cells in interpreting the data.2) Several redundancies and typos are still found in the manuscript, a careful reading will be needed before publication.

We have revised the manuscript according to your suggestion and to that of the reviewer#2.

We described the limitation of this study with the iPSC-derived MAIT cells in the discussion. Moreover, we have made edited the text by the professional English Editor so that there is no redundancy and typographical error.